# Analysis of mitochondrial m1A/G RNA modification reveals links to nuclear genetic variants and associated disease processes

Aminah Tasnim Ali[1], Youssef Idaghdour [2] & Alan Hodgkinson [1✉]

RNA modifications affect the stability and function of RNA species, regulating important downstream processes. Modification levels are often dynamic, varying between tissues and individuals, although it is not always clear what modulates this or what impact it has on biological systems. Here, we quantify variation in m1A/G RNA modification levels at functionally important positions in the human mitochondrial genome across 11,552 samples from 39 tissue/cell types and find that modification levels are associated with mitochondrial transcript processing. We identify links between mitochondrial RNA modification levels and genetic variants in the nuclear genome, including a missense mutation in *LONP1*, and find that genetic variants within *MRPP3* and *TRMT61B* are associated with RNA modification levels across a large number of tissues. Genetic variants linked to RNA modification levels are associated with multiple disease/disease-related phenotypes, including blood pressure, breast cancer and psoriasis, suggesting a role for mitochondrial RNA modification in complex disease.

[1] Department of Medical and Molecular Genetics, School of Basic and Medical Biosciences, King's College London, London SE1 9RT,, UK. [2] Program in Biology, Division of Science and Mathematics, New York University Abu Dhabi, PO Box 129188, Abu Dhabi, United Arab Emirates. ✉email: alan.hodgkinson@kcl.ac.uk

RNA modifications are post-transcriptional changes to the chemical composition of nucleic acids and represent a means by which RNA function can be fine-tuned[1]. Sites of RNA modification are often highly evolutionarily conserved and are crucial for processes such as development, cell signalling and maintenance of the circadian rhythm, pointing to a major role for RNA modification in fundamental cellular processes[1]. To date, over 160 different types of RNA modification have been identified[2], occurring on several types of RNA molecule, though they are found most abundantly on ribosomal and transfer RNAs[3]. The exact role of an RNA modification depends on the type, location and target of the modification. Within tRNAs, for example, modifications in the anticodon region can increase a tRNAs decoding capacity, and improve translational fidelity[4], whereas modifications to the core of a tRNA molecule can promote correct folding and structural stability[5]. Modifications to rRNA molecules are largely involved in the stabilisation of the ribosome structure, but can also facilitate protein synthesis[6], and modifications to mRNA molecules can affect the maturation, translation and degradation of an mRNA molecule by either recruiting additional proteins or by altering the secondary structure of the mRNA[7]. Importantly, not all modification levels are fixed; instead, some display a dynamic range of modification in different cell states and environments[8], which may in turn reflect a dynamic mode of RNA regulation.

Interest in RNA modifications has recently been renewed, due to the development of high-throughput technologies that can detect modifications on a transcriptome-wide scale. However, these studies often consider a small number of samples mostly limited to specific cell lines, and frequently focus on the detection of novel modification sites rather than attempt to survey the dynamic range of modification level across a large number of individuals[9–13]. In addition, several studies have used computational methods to quantify RNA modification levels using standard RNA sequencing (RNAseq) libraries[8,14–16]. Within this, we have previously shown that this approach is particularly effective for mitochondrial-encoded RNA, where due to its abundance in cells, we can detect levels of particular types of RNA modification (N1-methyladenosine and N1-methylguanine, m1A/G) at multiple functionally important positions along the mitochondrial transcriptome[14].

Mitochondria have essential roles in multiple cellular processes, including energy production, signalling, ion metabolism and apoptosis, and mutations in genes associated with mitochondrial processes have been linked to multiple different diseases[17–19]. The mitochondrial genome itself encodes 2 rRNA genes, 22 tRNA genes, and 13 mRNA genes[20], and is transcribed poly-cistronically before being processed according to the 'tRNA punctuation model'[21], whereby tRNAs interspersed between rRNA and mRNA genes form specific secondary degree structures that are used for recognition and cleavage by nuclear-encoded proteins to release individual mitochondrial RNA components[16,22–24]. Intermediate and mature RNA transcripts harbour extensive RNA modifications, which impact features such as transcript structure and stability that can be important for both processing and function[25]. Interestingly, steady state levels of mature mitochondrial transcripts vary substantially from the 1:1 ratio that might be expected from polycistronic transcription[26], indicating the importance of post-transcriptional processes in the maintenance of mitochondrial homoeostasis.

In illustration of this, knockdown of nuclear-encoded mitochondrial RNA processing enzymes in mice leads to the accumulation of unprocessed mitochondrial-encoded transcripts, decreased levels of protein synthesis and altered mitochondrial respiration rates[16,27]. Altered modification of mitochondrial-encoded RNA can have similar consequences (1) methylation (m1A/G) of bases at the ninth position of certain mitochondrial tRNAs (henceforth referred to as P9 sites) is understood to have an impact on the secondary structure and stability of the tRNA, influencing its ability to more permanently form the cloverleaf shape that is used for recognition and cleavage by nuclear-encoded proteins, thus impacting upon downstream levels of processed mitochondrial-encoded RNA[5,28]; (2) methylation (m1A) of mt-RNR2 transcripts at mtDNA position 2617 is believed to provide stabilising interactions to mature mitoribosomes, and lack of RNA methylation at this position has been linked to impaired mitochondrial protein synthesis[29]; (3) methylation (m1A) of mt-ND5 transcripts at mtDNA position 13710 varies according to tissue type[11], and interferes with translation through mitoribosome stalling and leads to decreased protein levels[9]. In this study, we focus on quantifying variation in mitochondrial-encoded RNA methylation levels at these three classes of site on a population level in 11,552 RNA sequencing samples across 39 different tissue/cell types. We perform quantitative trait mapping using mitochondrial-encoded RNA methylation rates in order to identify nuclear genetic variants and genes that are involved in the regulation of these processes across tissues, and to unravel their downstream functional consequences.

## Results

**Overview.** In order to study variation in methylation (m1A/G) levels of mitochondrial-encoded RNA across multiple tissue types, we mapped and filtered 13,857 RNAseq samples from 39 tissue/cell types, across 5 independent datasets (see Methods, Supplementary Table 1), to the human reference genome using a stringent pipeline optimised for the analysis of mitochondrial data. After quality filtering, 11,552 samples remained for analysis. It has previously been shown that the levels of m1A/G modifications can be inferred at particular positions using the proportion of mismatching bases in RNAseq data[8,16,26]. The assumption behind this approach is that chemical modifications of RNA act to either block the reverse-transcription enzyme during RNA sequencing library preparation or cause the enzyme to mis-incorporate nucleotides resulting in mismatched alleles when compared with the reference nucleotide. The proportion of mismatches observed at modified sites has been shown to be systematic and repeatable across experiments (with mismatched alleles not present in the corresponding DNA)[14], and the quantitative nature of using the proportion of mismatches as a proxy for RNA methylation rate was demonstrated by looking at primer extension rates at sites known to be modified[10] (see Methods). As such, we use this proportion to quantify the level of RNA methylation at three categories of modified site where RNA methylation is known to be functionally important (see above); (1a) methylation at P9 sites of 13 of the 22 different mt-tRNAs along the mitochondrial genome where methylation levels are high enough to be detected via this approach, (1b) an average estimate of methylation level across 11 different mt-tRNA P9 sites that consistently show variation in whole blood[14], under the assumption that m1A/G levels may be driven by the same shared mechanism along the mitochondrial transcriptome, (2) methylation at mt-DNA position 2617 within mt-RNR2 and (3) methylation at mt-DNA position 13170 within mt-ND5.

**RNA methylation patterns across tissues.** Across the 39 different tissue/cell types examined, whole blood, brain, muscle and nerve tissues show the highest levels of inferred RNA methylation across all tRNA P9 sites combined, with average levels of 11–25%, 7–12%, 11%, and 10%, respectively (ranges shown where multiple dataset-tissue type pairs are available). In contrast, the lowest levels of inferred tRNA methylation are observed in cell lines,

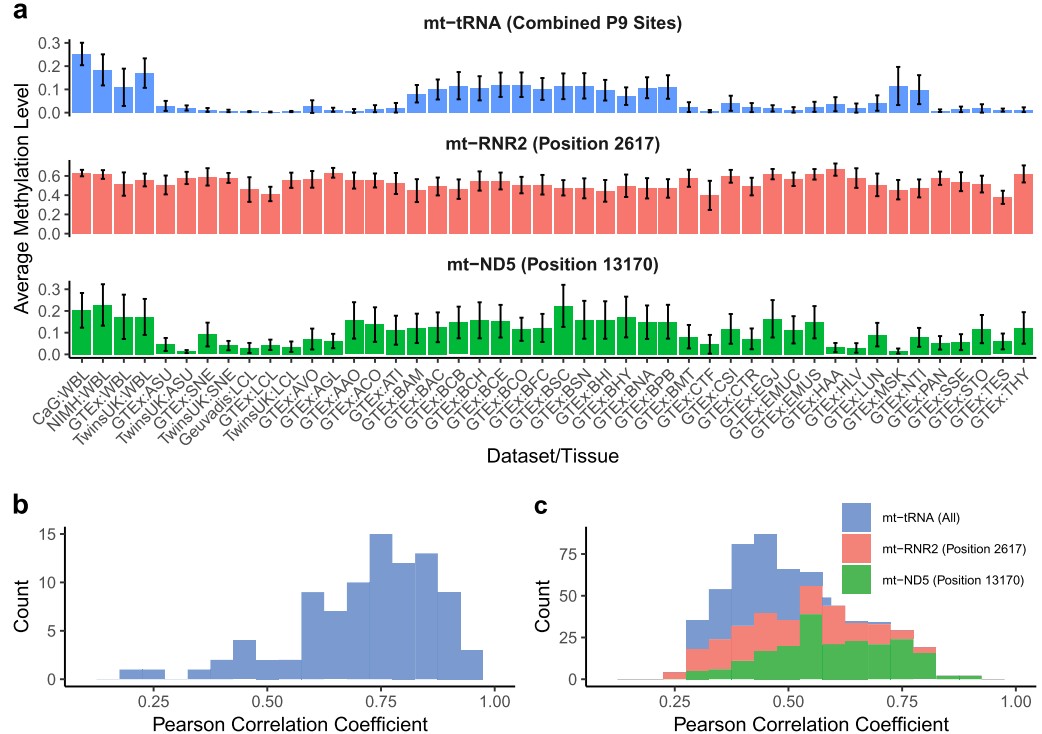

**Fig. 1 Inferred mitochondrial RNA methylation levels. a** Average RNA methylation levels calculated across datasets and tissue/cell types at three categories of methylated site: averaged values across 11 mt-tRNA P9 sites, mt-RNR2 (position 2617) and mt-ND5 (position 13170). Bars show 1 standard deviation from the mean. **b** Correlation coefficients between inferred tRNA P9 methylation levels within an individual, measured across individuals in all datasets and tissue types (Pearson correlation, a total of 91 comparisons made between p9 sites using data from $N = 11,552$ individuals). **c** Correlation coefficients between samples with inferred methylation levels measured in multiple tissues, measured at tRNA P9 sites, mt-RNR2 and mt-ND5. Data shown for significant correlations only ($P < 0.05$ after Bonferroni correction, Pearson correlation). WBL whole blood, ASU adipose subcutaneous, SNE skin not sun exposed, LCL lymphoblastoid cell line, AVO adipose visceral omentum, AGL adrenal gland, AAO artery aorta, ACO artery coronary, ATI artery tibial, BAM brain amygdala, BAC brain anterior cingulate cortex, BCB brain caudate basal ganglia, BCH brain cerebellar hemisphere, BCE brain cerebellum, BCO brain cortex, BFC brain frontal cortex, BSC brain spinal cord cervical, BSN brain substantia nigra, BHI brain hippocampus, BHY brain hypothalamus, BNA brain nucleus accumbens basal ganglia, BPB brain putamen basal ganglia, BMT breast mammary tissue, CTF cells transformed fibroblasts, CSI colon sigmoid, CTR colon transverse, EGJ esophagus gastroesophageal junction, EMUC esophagus mucosa, EMUS esophagus muscularis, HAA heart atrial appendage, HLV heart left ventricle, LUN lung, MSK muscle skeletal, NTI nerve tibial, PAN pancreas, SSE skin sun exposed, STO stomach, TES testis, THY thyroid.

with average levels across P9 sites ranging between 0.3–0.5% in LCLs and 0.7% in transformed fibroblasts (Fig. 1a, Supplementary Fig. 1). Inferred RNA methylation levels also vary between individual tRNA P9 positions along the mitochondrial genome. For example, across all datasets inferred methylation levels at tRNA position 3238 have an average value of 0.9%, whereas the average methylation levels at position 8303 is 12%. To test whether mt-tRNA P9 methylation levels are similar between different P9 sites along the mitochondrial transcriptome within an individual, we measured correlations between inferred methylation levels at each pair of mt-tRNA P9 sites within each dataset-tissue type pair. Across all individuals and dataset-tissue types, all correlation coefficients were significant after Bonferroni correction ($P < 0.05$, Pearson correlation), and centred around 0.75 (ranging between 0.18 and 0.95, Fig. 1b), suggesting that methylation levels at different P9 sites are broadly consistent along the mitochondrial transcriptome within each individual, and thus may be driven by shared mechanisms.

Outside of tRNAs, the average inferred methylation level at *mt-rRNR2* (position 2617) and *mt-ND5* (position 13710) transcripts are generally high across all tissues examined (Fig. 1a), with sample wide average values of 55% and 10%, respectively. Average inferred *mt-RNR2* transcript methylation levels range between 38% in GTEx Testis and 67% in GTEx Heart (Atrial Appendage), and average levels of inferred transcript methylation within *mt-ND5* range between 1% in subcutaneous adipose data

from TwinsUK and 23% in whole blood data from the NIMH. Within dataset-tissue pairs, there is considerable variation in inferred transcript methylation levels across individuals (Supplementary Fig. 1); for example, at position 2617 in the CARTaGENE whole blood dataset, inferred methylation levels vary between 0.48 and 0.72, and between 0.11 and 0.76 in the Geuvadis lymphoblastoid cell line (LCL) data.

To test if mitochondrial RNA methylation levels (at tRNA P9 sites, at the *mt-rRNR2* site and at the *mt-ND5* site) are correlated across tissue types within an individual, we selected individuals from the GTEx dataset, where RNAseq data from the largest number of alternative tissue types were available, and measured pairwise correlations. For tRNA P9 sites, 11% of pairwise comparisons were significant after Bonferroni correction ($P < 0.05$, Pearson correlation. For significant correlations: median $r = 0.42$, range 0.28–0.7, Fig. 1c). At the *mt-RNR2* (position 2617) and *mt-ND5* (position 13710) sites, 76% (for significant correlations: median $r = 0.49$, range 0.24–0.81) and 95% (for significant correlations: median $r = 0.59$, range 0.3–0.88) of correlation coefficients are significant after Bonferroni correction ($P < 0.05$, Pearson correlations) (Fig. 1c). Collectively, these results demonstrate detectable variation in mitochondrial-encoded RNA methylation levels at the individual and population level, as well as consistency in the levels observed along the mitochondrial transcriptome and across tissues, suggesting the presence of shared underlying regulatory mechanisms.

**Table 1 Significant associations between nuclear genetic variation and inferred mitochondrial RNA methylation level for meta-analysed tissues.**

| Tissue | Mito position | rsID | CHR | BP | A1 | N | BETA | P | SNP type | SNP location | Mediator genes |
|---|---|---|---|---|---|---|---|---|---|---|---|
| Whole blood | 585 | rs13874 | 3 | 66419956 | T | 3 | 0.0204 | 3.36E−51 | Missense | SLC25A26 | SLC25A26 |
| Whole blood | 585 | rs1475041 | 14 | 35793550 | G | 4 | 0.0138 | 6.21E−21 | Intergenic | NA | – |
| Whole blood | 1610 | rs13874 | 3 | 66419956 | T | 3 | 0.0453 | 1.50E−65 | Missense | SLC25A26 | SLC25A26 |
| Whole blood | 1610 | rs11156878 | 14 | 35735967 | G | 4 | 0.042 | 4.64E−35 | Missense | MRPP3 | – |
| Whole blood | 3238 | rs13874 | 3 | 66419956 | T | 3 | 0.002 | 8.77E−15 | Missense | SLC25A26 | – |
| Whole blood | 3238 | rs61988267 | 14 | 35730799 | T | 4 | 0.003 | 5.05E−20 | Intronic | MRPP3 | – |
| Whole blood | 4271 | rs69088276 | 14 | 35769451 | A | 4 | 0.0086 | 1.15E−14 | Intronic | PSMA6 | – |
| Whole blood | 5520 | rs11156878 | 14 | 35735967 | G | 4 | 0.0477 | 3.23E−37 | Missense | MRPP3 | – |
| Whole blood | 7526 | rs1084535 | 3 | 66360030 | A | 3 | 0.0241 | 4.90E−18 | Intronic | SLC25A26 | SLC25A26 |
| Whole blood | 7526 | rs140678103 | 14 | 35765525 | G | 3 | 0.0335 | 2.36E−21 | Intronic | PSMA6 | – |
| Whole blood | 8303 | rs74422990 | 14 | 35726093 | G | 3 | 0.0501 | 6.19E−29 | Intronic | MRPP3 | – |
| Whole blood | 9999 | rs11156878 | 14 | 35735967 | G | 4 | 0.0757 | 1.69E−87 | Missense | MRPP3 | – |
| Whole blood | 9999 | rs11085147 | 19 | 5711930 | T | 4 | −0.0431 | 4.68E−15 | Missense | LONP1 | – |
| Whole blood | 10413 | rs3820190 | 1 | 12033120 | C | 3 | −0.0241 | 3.11E−12 | Intronic | PLOD1 | PLOD1 |
| Whole blood | 10413 | rs11156878 | 14 | 35735967 | G | 4 | 0.082 | 1.10E−87 | Missense | MRPP3 | – |
| Whole blood | 12146 | rs11156878 | 14 | 35735967 | G | 4 | 0.0659 | 8.94E−81 | Missense | MRPP3 | – |
| Whole blood | 12146 | rs11085147 | 19 | 5711930 | T | 4 | −0.0335 | 1.27E−11 | Missense | LONP1 | – |
| Whole blood | 12274 | rs11156878 | 14 | 35735967 | G | 4 | 0.044 | 4.32E−69 | Missense | MRPP3 | – |
| Whole blood | 14734 | rs11156878 | 14 | 35735967 | G | 4 | 0.0085 | 1.03E−22 | Missense | MRPP3 | – |
| Whole blood | 15896 | rs11156878 | 14 | 35735967 | G | 4 | 0.0115 | 1.45E−55 | Missense | MRPP3 | PPP2R3C, MRPP3 |
| Whole blood | Averaged tRNA P9 | rs11156878 | 14 | 35735967 | G | 4 | 0.0459 | 1.58E−116 | Missense | MRPP3 | – |
| Whole blood | 2617 | rs11684695 | 2 | 29088450 | T | 4 | 0.0233 | 5.50E−99 | Intronic | TRMT61B | PPP1CB, TRMT61B, CLIP4 |
| Whole blood | 2617 | rs2627773 | 2 | 55900459 | A | 4 | −0.0083 | 1.82E−12 | Intronic | PNPT1 | PNPT1 |
| Whole blood | 2617 | rs13874 | 3 | 66419956 | T | 3 | 0.0093 | 2.22E−14 | Missense | SLC25A26 | – |
| Whole blood | 13710 | rs10826790 | 10 | 30643872 | G | 4 | 0.0284 | 3.34E−28 | Intergenic | NA | – |
| Adipose | 10413 | rs200541481 | 14 | 35761028 | TCA | 2 | 0.0051 | 4.14E−13 | Intronic | PSMA6 | – |
| Adipose | Averaged tRNA P9 | rs11156878 | 14 | 35735967 | G | 2 | 0.005 | 4.32E−14 | Missense | MRPP3 | – |
| Adipose | 2617 | rs10166861 | 2 | 29061111 | A | 2 | 0.0471 | 3.58E−80 | Intronic | SPDYA | TRMT61B |
| Adipose | 13710 | rs2247084 | 10 | 30620625 | G | 2 | 0.0031 | 6.54E−14 | Intronic | MTPAP | – |
| Skin | 2617 | rs10865508 | 2 | 29053704 | C | 2 | 0.0363 | 2.91E−67 | Intronic | SPDYA | – |
| Skin | 13710 | rs2689214 | 10 | 30632228 | A | 2 | 0.0106 | 1.82E−17 | Intronic | MTPAP | – |
| LCLs | 2617 | rs55785599 | 2 | 29087814 | A | 3 | 0.0349 | 1.13E−51 | Intronic | TRMT61B | – |

Alleles presented in the 'A1' column represent the minor allele, and the values in column 'N' represent the number of studies contributing the meta-analysis, for that row.

**Quantitative trait mapping**. To identify nuclear genetic variation associated with inferred mitochondrial-encoded RNA methylation levels, we obtained genome-wide genotyping data for the same samples that we had RNA data for, and carried out quantitative trait mapping within each of the 39 tissue/cell types for the level of methylation at functionally important positions within the mitochondrial genome. This included inferred methylation levels at (1a) 13 different tRNA P9 sites along the mitochondrial genome, (1b) an average measure across multiple tRNA P9 sites (see Methods), (2) position 2617 within *mt-rRNR2* and (3) at position 13710 within *mt-ND5*. For tissues where we had multiple independent datasets, which includes whole blood, subcutaneous adipose, skin (non-sun exposed) and LCLs, we then carried out meta-analyses. Across all datasets, we corrected for multiple testing by accounting for genome-wide testing, the number of methylation sites examined and the number of tissues included in the analysis, resulting in a significance threshold of $P < 6.79 \times 10^{-11}$.

Across all tissue types and mitochondrial RNA positions where we quantify methylation levels, we find a total of 47 significant associations (peak nuclear genetic variant and mitochondria encoded RNA methylation level pairs). Most associations occur in tissues for which we have multiple independent datasets, and thus larger sample sizes (Table 1, Fig. 2); 25 nuclear genetic loci are significantly associated with inferred mitochondrial-encoded RNA methylation levels in whole blood, 4 are detected in subcutaneous adipose, 2 in non-sun exposed skin and 1 in LCLs. In single dataset tissues, we identify 15 significantly associated genomic regions across multiple different tissue types: adipose (visceral omentum), artery (aorta and tibial), nerve (tibial), oesophagus (muscularis and gastroesophageal junction), transformed cells (fibroblasts) and skin (sun exposed) (Supplementary Table 2). In total, 17 peak nuclear

genetic variants fall at missense sites, 22 within introns and 8 in intergenic regions. Across all associated loci, many regions are overlapping in different tissue types and methylation positions; removing all regions that overlap leaves a total of 3 unique regions on the nuclear genome associated with inferred methylation levels at mitochondrial tRNA P9 sites, 2 unique regions associated with inferred methylation levels at *mt-RNR2*, 1 unique region associated with inferred methylation levels at both tRNA P9 sites and the *mt-RNR2* site and 1 unique region associated with *mt-ND5*.

To further consider the underlying genetic architecture of variable mitochondrial RNA methylation levels across individuals, for sites in mitochondrial RNA where we identify two independent nuclear loci associated with methylation levels with the same direction of effect, we tested whether independent alleles have an additive effect. Under these criteria we consider five methylated positions in whole blood data, and in all of these cases, we observe a significant change in inferred methylation levels associated with carrying two effect alleles (one at each independent loci) versus carrying only one (Fig. 3, $P < 0.05$ after Bonferroni correction, t-tests).

**Functional annotation of nuclear genetic variation**. In order to identify the potential genes and mechanisms through which nuclear genetic variants are associated with mitochondrial-encoded RNA methylation levels, we tested whether the peak variant in each region was either a missense variant (and therefore potentially functional) or acted via modulation of the expression of a nearby nuclear gene in *cis*, which then influences mitochondrial RNA methylation levels, by performing mediation analysis (requiring an association between the expression of a nearby nuclear-encoded gene and the peak nuclear genetic

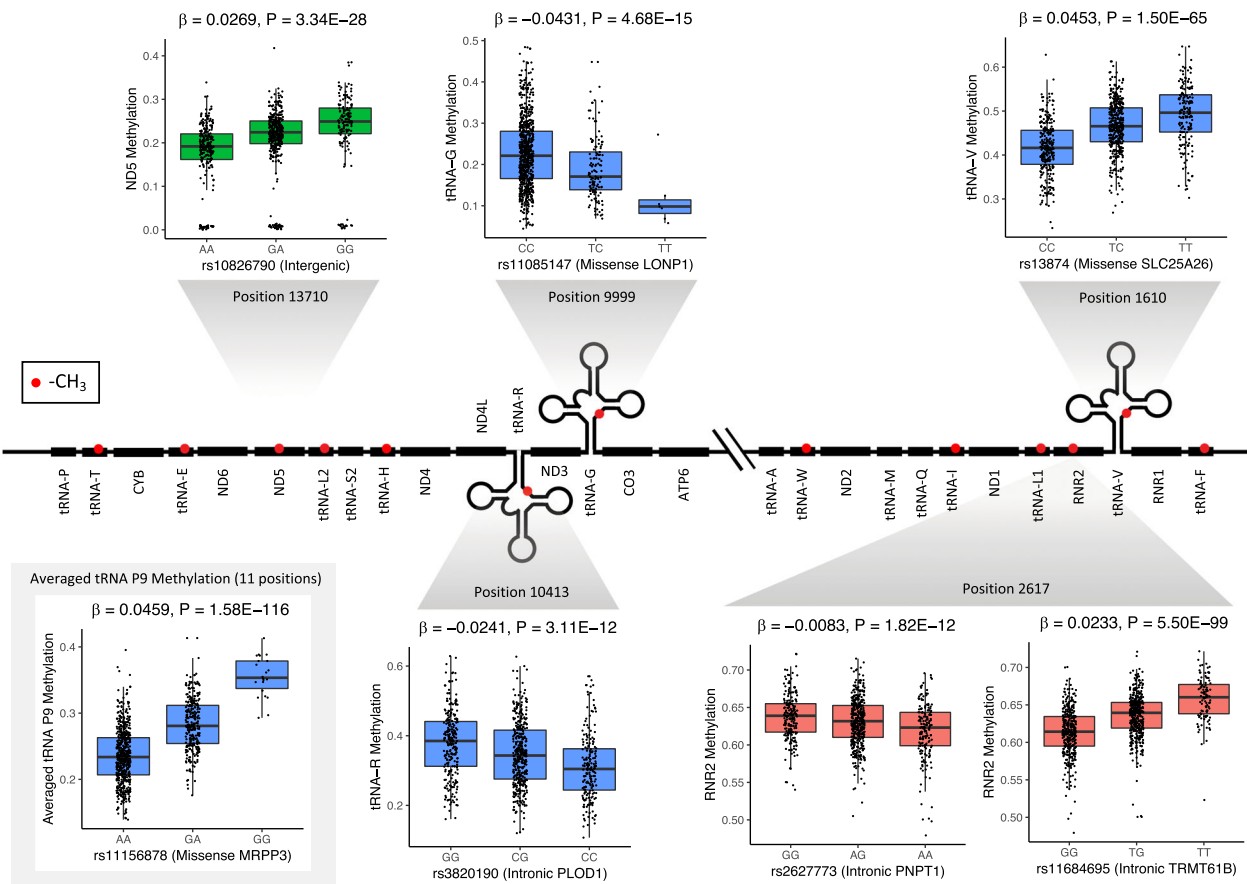

**Fig. 2 Relationship between genotype and inferred methylation level at multiple positions on the nuclear genome and mitochondrial transcriptome, respectively.** Methyl groups are represented by red circles along the mitochondrial transcriptome, and inferred methylation levels are shown at three categories of methylated site: at tRNA P9 sites (blue), at position 2617 within *mt-RNR2* transcripts (red) and at position 13170 within *mt-ND5* transcripts (green). Averaged inferred levels of methylation across 11 mt-tRNA P9 sites are additionally shown in the grey shaded box (bottom left). Beta estimates and *P*-values displayed are from meta-analysis of four independent whole blood datasets (sample sizes shown in Supplementary Table 1), and methylation levels and genotypes displayed in boxplots originate from the CARTaGENE dataset.

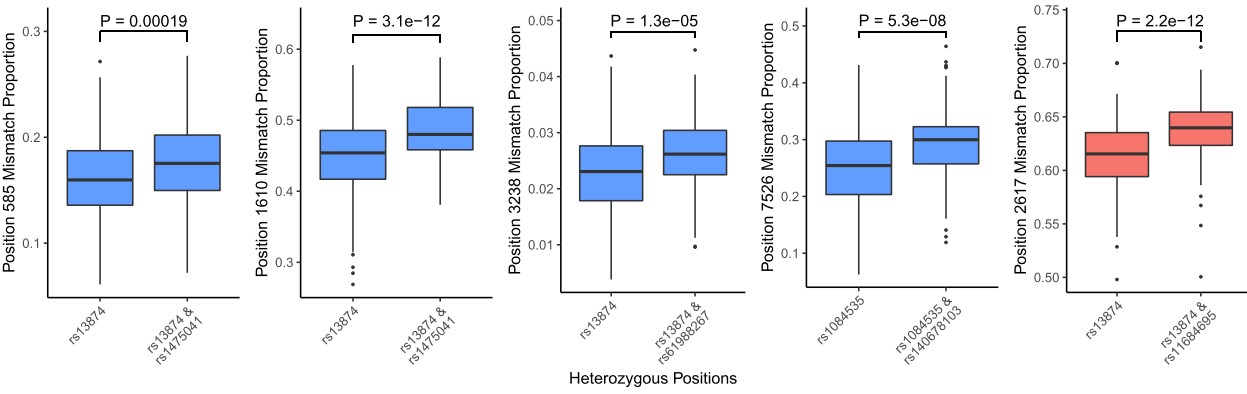

**Fig. 3 Additive effects of independent nuclear genetic variants associated with inferred RNA methylation levels in whole blood data.** In each case, individuals that are heterozygous for the first genetic variant (and homozygous for the reference allele at the second variant) are compared with individuals that are heterozygous at both sites (*t*-test, data from the CARTaGENE project, from left to right: N = 373, 377, 377, 360, 333 independent biological samples).

variant associated with inferred mitochondrial RNA methylation, and then significant mediation of the initial association via bootstrapping, requiring an average causal mediation effect with $P < 0.05$, corrected for the number of tests).

Applying this approach, we identify a number of novel candidate causal genes (to our knowledge) that may be involved in modulating mitochondrial tRNA methylation levels. First, we

find a missense variant (rs11085147) in *LONP1* that is associated with inferred methylation levels at *mt-tRNA-G* (position 9999) and *mt-tRNA-H* (position 12,146). *LONP1* codes for a mitochondrial matrix protein that is involved in degradation of damaged or unfolded polypeptides, in addition to the maturation of certain mitochondrial proteins[30], and is also thought to directly bind to mitochondrial DNA and RNA[31]. Second, an intronic variant

**Table 2 Regression analyses between methylation level at mitochondrial tRNA P9 sites and expression of its 5′ mRNA, in whole blood datasets.**

| Mitochondria position | 5′ mRNA | Mitochondrial tRNA | CARTaGENE Beta coefficient | CARTaGENE P | GTEx Beta coefficient | GTEx P |
|---|---|---|---|---|---|---|
| 1610 | MT-RNR1 | V | 0.2201271 | 3.16E−07 | 0.09739377 | 0.4027136 |
| 3238 | MT-RNR2 | L1 | 2.361507 | 1.31E−12 | 2.511323 | 2.16E−05 |
| 4271 | MT-ND1 | I | 0.1560262 | 0.1780689 | 0.2212783 | 0.1716995 |
| 5520 | MT-ND2 | W | 0.07553068 | 0.02423747 | 0.03422895 | 0.6816734 |
| 8303 | MT-CO2 | K | 0.04620692 | 0.01972715 | 0.005807783 | 0.9465179 |
| 9999 | MT-CO3 | G | −0.0476643 | 0.02866581 | −0.1192088 | 0.04399886 |
| 10413 | MT-ND3 | R | 0.09531144 | 0.000149767 | 0.05458773 | 0.31342 |
| 12146 | MT-ND4 | H | 0.01657878 | 0.4552959 | 0.1370204 | 0.05093543 |
| 15896 | MT-CYB | T | 0.02015266 | 0.8987243 | 0.2027844 | 0.2833017 |

(rs3820190) in *PLOD1* that is associated with inferred mitochondrial RNA methylation levels at *mt-tRNA-R* (position 10413), is also associated with the expression of *PLOD1* in *cis*, which in turn shows evidence of significant mediation of the association between rs3820190 and inferred mitochondrial RNA methylation (Table 1). *PLOD1* is not currently thought to be involved in mitochondrial function, however, the expression of *MFN2* (a gene involved in mitochondrial fusion) is also associated with rs3820190, and although it was not significant in mediation analysis, it remains a viable candidate for the causal gene in this case due to its involvement in mitochondrial processes. Third, on top of replicating previously identified links between missense variants rs13874 and rs11156878 (within *SLC25A26* and *MRPP3*, respectively) and mitochondrial-encoded tRNA methylation levels using larger sample sizes in whole blood[14], we also detect significant associations between rs11156878 and inferred tRNA P9 methylation levels in subcutaneous adipose and nerve tissues (Table 1 and Supplementary Table 2), showing that these links may be important in many regions across the body. *MRPP3* is part of a complex responsible for 5′ mt-tRNA processing, and thus is directly involved in processes that may impact RNA methylation.

Outside of mitochondrial tRNAs, we identify an association between an intronic variant (rs2627773) that is significantly associated with the expression of *PNPT1*, which in turn significantly mediates the original association between the genetic variant and inferred mitochondrial RNA methylation levels at position 2617 within *mt-RNR2* in whole blood. *PNPT1* codes for an RNA binding protein that plays a role in various different RNA metabolic processes, and thus may modulate RNA methylation either directly or through interactions with other proteins. Finally, we identify intronic variants (rs11684695 and rs10166861) that are significantly associated with inferred methylation levels at position 2617 within *mt-RNR2* (in whole blood and subcutaneous adipose, respectively) that mediate these associations through the expression of genes including *TRMT61B*, a mitochondrial methyltransferase. The association in whole blood was originally identified in a previous study[14], but the link between *TRMT61B* and RNA methylation within *mt-RNR2* in subcutaneous adipose we identify here suggests that this gene may play a more global role in the regulation of RNA modification.

**Cross-tissue analysis**. As whole blood made up our largest dataset (2424 RNAseq libraries across four datasets), we tested whether nuclear genetic variants associated with inferred mitochondrial RNA methylation levels in whole blood operate in a tissue specific or tissue-wide manner. For each of the 25 significant mitochondrial methylation site-variant pairs in whole blood, we tested for evidence of replication (correcting for the number of methylation site-variant pairs), with the same direction of effect in all other tissues. In total, 17/25 position-SNP pair associations replicate in at least one other tissue type (Supplementary Table 3). rs11156878, which is associated with the average level of inferred RNA methylation at tRNA P9 sites, replicates in 22 additional tissues. rs11684695, associated with inferred RNA methylation levels of position 2617, showed replication in 25 other tissues, and rs10826790, associated with RNA methylation at position 13170, is replicated in 6 additional tissues (Supplementary Table 3). This suggests that certain genetic loci associated with mitochondrial RNA methylation levels in whole blood are active in multiple other tissue types, and potentially in a system-wide manner. Other variants, however, such as rs13874, which associated with inferred methylation levels at multiple individual mt-tRNA P9 sites and within *mtRNR2* in whole blood, does not show evidence of association in other tissues, suggesting that it may either be tissue specific, or that there is insufficient power to detect the association at smaller sample sizes.

**Consequences of variation in mitochondrial RNA methylation**. Methylation modifications at mt-tRNA P9 sites are thought to stabilise the secondary structure of the corresponding mt-tRNA sequences within the mitochondrial transcriptome[28]. Since tRNA structure is important for post-transcriptional substrate recognition and cleavage, we tested if inferred methylation levels at tRNA P9 sites were related to mt-mRNA expression of the adjacent gene for the 9 occurrences where an mRNA or rRNA gene is found immediately upstream of a tRNA. Using data from the CARTaGENE project, which is the largest, paired-end dataset, we find that inferred tRNA methylation levels are significantly associated with the expression levels of genes immediately upstream in 6 out of 9 cases at $P < 0.05$ (5 of which show positive relationships, linear regressions) and 3/6 remain significant after correcting for the number of pairs tested (Table 2). To test for replication of the relationship, we used the GTEx whole blood dataset, which is the second largest, paired-end dataset of unrelated samples. We find that 2/6 nominally significant relationships show evidence of replication, with the same direction of effect ($P < 0.05$ after Bonferroni correction, linear regressions). These include the positive association between inferred methylation levels within *mt-tRNA-L1* and the expression of *mt-RNR2*, and the negative relationship between inferred methylation levels within *mt-tRNA-G* and the expression of *mt-CO3* (Table 2).

**Overlap with disease associated loci**. Finally, to identify potential links between genetic variants associated with mitochondrial RNA methylation levels and complex traits and phenotypes, we looked for overlaps between peak associated nuclear variants

(and SNPs in LD, $D' \geq 0.9$), and genome-wide significant disease associated variants in the NHGRI-EBI GWAS Catalogue[32]. We find overlaps with blood pressure and atrial fibrillation, glaucoma and intraocular pressure, breast cancer and psoriasis (Supplementary Table 4).

First, the intronic variant in PNPT1 (rs2627773) that is associated with inferred methylation levels of mt-RNR2 (position 2617), is in LD with rs1975487, which is associated with diastolic blood pressure[33]. Mitochondria have previously been linked to increased blood pressure, predominantly through mechanisms involving mitochondrial oxidative stress, however, the exact mechanisms by which this is the case remain unclear[34]. We have previously observed associations between nuclear genetic variants influencing the expression of certain mitochondrial genes, mediated through the expression of PNPT1[35], and the identification of the overlap here suggests that RNA methylation level may also be playing a contributory role in the process. Furthermore, nuclear genetic variants (rs13874, rs1084535) associated with inferred methylation levels at multiple mt-tRNA P9 sites and at mt-RNR2 are in LD with rs34080181, which has been associated with atrial fibrillation[36]. High blood pressure is a risk factor for atrial fibrillation[37], however, mitochondrial dysfunction itself has been implicated with the development of atrial fibrillation, through the generation of reactive oxidative species and the alteration of calcium homeostasis and oxygen consumption[38].

Second, genetic variants (rs10166861, rs10865508, rs34611659 and rs13033423) associated with inferred methylation levels at mt-RNR2 in multiple tissue types are in LD with rs4577244, which has been linked to breast cancer[39]. The role of mitochondria in cancer has been debated since the discovery of the Warburg effect[40], and subsequent research has linked many additional pathways/features of the mitochondria to tumorigenesis, including through alterations in its roles in cell death, metabolism, and oxidative stress[41]. In previous work, we observed an increase in the level of tRNA P9 methylation levels in cancer tumours vs matched normal tissues[15]; our observation of an overlap here suggests that methylation levels at mt-RNR2 may also be involved.

Third, rs2627773, associated with inferred methylation levels in mt-RNR2 in whole blood, is in LD with rs2627761, which has been associated with glaucoma[42]. Age related mitochondrial dysfunction has been suggested to play a role in development of glaucoma, through lack of energy availability for repair mechanisms in the eyes[43]. Similarly, genetic variants (rs11685682, rs34611659 and rs4132617) associated with inferred methylation levels at mt-RNR2 are in LD with rs147972440, which has been linked to intraocular pressure[44]. Intraocular pressure is a risk factor for glaucoma, suggesting that the mitochondria may additionally be involved in the development of glaucoma through its impact on intraocular pressure.

Finally, nuclear genetic variants (rs1475041, rs11156878, rs61988267, rs74422990 and rs200541481) associated with inferred methylation levels at multiple different mt-tRNA P9 sites are in LD with rs8016947, which has been associated with psoriasis[45]. Increased levels of mtDNA have been observed in the serum of psoriasis patients compared with controls, and mitochondria have been suggested to be involved in psoriasis by triggering inflammation through reduced mitochondrial apoptosis and extracellular leak of mitochondrial DNA, eliciting an immune response[46].

## Discussion

RNA modifications represent an additional layer of control in the regulation of gene expression. They are found extensively throughout both the nuclear and mitochondrial transcriptome, where they play important roles in structural stability and translation efficiency. Using mitochondria as a model system, we characterise RNA methylation (m1A/G) levels (via RNA sequencing mismatch proportions) at multiple functionally important sites on the mitochondrial transcriptome, across a total of 39 tissues/cell types. We find that RNA methylation levels are correlated along the transcriptome, but vary between tissues, with blood and brain tissues showing the highest levels of variation. As the mitochondrial and nuclear genomes have co-evolved over evolutionary time, we also link variation in mitochondrial RNA methylation levels to genetic variation in the nuclear genome.

In total, we associate 6 nuclear genes to fundamental biological processes taking place in human mitochondria. Within this we identify, to our knowledge, novel associations between mitochondrial RNA methylation levels and a missense mutation in LONP1, as well as independent non-coding variants that may be operating through modulating the expression of PLOD1 and PNPT1. Furthermore, we find that previously identified associations[14], which have been linked to MRPP3 and TRMT61B, occur in multiple tissue types, which may have important implications for disease.

MRPP3 codes for the catalytic subunit of mitochondrial RNase P, a complex responsible for the 5′ cleavage of mt-tRNAs[22], and is active only in the presence of its other subunits[47], MRPP1 and MRPP2. The MRPP1 and MRPP2 sub-complex, however, is able to carry out its methyltransferase activity independently of MRPP3[48], so the association between rs11156878 within MRPP3 with methylation of mt-tRNAs is likely detected due to its effect on cleavage capacity. TRMT61B is a methyltransferase that is responsible for the methylation of position 58 on certain mitochondrial tRNAs, in addition to position 2617 in mt-RNR2[29]. In the present study, an intronic SNP is associated with increased levels of methylation at mt-RNR2, as well as increased expression of TRMT61B, likely explaining the relationship detected here. Of genes newly implicated with mitochondrial RNA methylation levels, LONP1 has been shown to degrade MRPP3 as part of the mitochondrial unfolded protein response[49], possibly explaining its association with methylation level, or alternatively it may influence methylation levels more directly as it is known to bind to mitochondrial DNA and RNA. Finally, PNPT1 is involved in multiple metabolic RNA processes in mitochondria, and reduction of PNPT1 levels results in impaired mitochondrial processing and accumulation of large polycistronic transcripts, possibly due to its connection to the import of RNase P RNA into the mitochondria[50,51], again likely explaining why it is associated with methylation level in this study.

Interestingly, rs13874, a missense mutation in SLC25A26, tends to only be associated with tRNAs towards the beginning of the mitochondrial transcript in whole blood, and does not show evidence of replicating across tissues. SLC25A26 is a mitochondrial carrier protein that is responsible for transporting S-adenosylmethionine into the mitochondria, which is a methyl group donor in methylation reactions. The absence of rs13874 replication across tissues may be related to the fact that the highest levels of methylation are seen in whole blood, in combination with SLC25A26 concurrently having the lowest levels of expression in whole blood (https://gtexportal.org/home/gene/SLC25A26). Therefore, the effect of a possible transportation deficiency may only be observed in tissues where the requirement for methylation is high. As blood is the tissue in which we detect the highest levels of methylation, but also the lowest levels of SLC25A26 expression, this mutation may have particularly important implications for blood-based processes and diseases.

The downstream functional consequences of altered mitochondrial RNA processing are well documented in human cell lines and model organisms[16,52], but here using in vivo data we

show that natural variation of mitochondrial RNA methylation levels in 'healthy' individuals may influence mitochondrial processes (namely changes in mitochondrial gene expression). Disruption or perturbation of the function of nuclear genes that we have implicated in mitochondrial RNA methylation can have serious phenotypic consequences. In humans, for example, a mutation in *PNPT1* has also been linked with impaired import of RNA into the mitochondria, and leads to combined oxidative phosphorylation deficiency[53] and also autosomal recessive deafness[51]; missense mutations in *LONP1* have been implicated in CODAS syndrome, which is a developmental disorder affecting multiple systems (cerebral, ocular, dental, auricular and skeletal)[54] and mutations in *SLC25A26* have been linked to combined oxidative phosphorylation deficiency[55]. Similarly, *TRMT61B* has been identified as a differentially expressed gene in a small cohort of Alzheimer's disease cases, when compared with matched controls[52], and knockdown of the drosophila homologue of *MRPP3* leads to the loss of locomotive function in *Drosophila*, similarly to what is seen in Parkinson's disease[27].

These examples are phenotypically varied and are the result of extreme alterations in the function of the corresponding gene. The genetic variation associated with RNA methylation levels in this study have less extreme effects, however, they are linked with changes in post-transcriptional processing and downstream expression of mitochondrial genes. While this variation might be tolerated under normal physiological situations, the introduction of stressful situations, for example, during increased mitochondrial damage through ageing, may be enough to push a tissue that is heavily reliant on appropriate mitochondrial function into dysfunction. Conversely, altered mitochondrial expression over a long duration may be enough to lead to negative later life consequences. Impaired mitochondrial gene expression due to the heterozygous knockout of *PTCD1*, a mitochondrial RNA processing enzyme, for example, has been linked to later-life obesity in mice[52]. In this study, we find overlaps between genetic variants associated with mitochondrial RNA methylation levels and variants linked to blood pressure and atrial fibrillation, glaucoma and intraocular pressure, breast cancer and psoriasis, suggesting that altered mitochondrial RNA modification may play a role in complex diseases. Overall, it will be important to fully disentangle the genetic and molecular mechanisms underlying post-transcriptional processes in the mitochondria across a range of both healthy and diseased states, building on the population level framework we describe here, since changes in these processes may dramatically alter mitochondrial function in a multitude of cellular environments.

Finally, our approach makes use of sequence mismatches in RNA sequencing data as a proxy for m1A/G RNA modification levels. This approach is semi-quantitative by nature since misincorporation of nucleotides by the reverse-transcription enzyme at modified sites during RNA sequencing library preparation is not infallible (see Methods). However, our approach allows us to survey rates of RNA modification across a large number of individuals and tissues, drawing power to make important biological inferences from these data. It is hoped in the future that other high-throughput approaches (such as Nanopore sequencing or site-specific mass spectrometry) may have the potential to more directly quantify RNA modification levels across a wider range of modification types, and thus allow for an expanded view on genetic drivers and downstream consequences of dynamic RNA modification regulation.

## Methods

**Data description**. RNA sequence and genotype data were obtained from five independent, publicly available projects, including:

CARTaGENE[56]: A population-based cohort comprised of people aged between 40 and 69, from Quebec, Canada. Whole blood samples were taken for RNA sequencing and genotyping, producing 100 bp paired-end RNAseq reads and genotypes from the Illumina Omni2.5M genotyping array. Samples with RNAseq data from multiple sequencing runs, that passed quality control, were merged before the alignment stage.

NIMH (National Institute of Mental Health) Genomics Resource[57]: Whole blood samples were collected for RNA sequencing and genotyping from the Depression Genes and Networks study. Individuals were aged 21–60, and are a case/control cohort. 50 bp single-end RNAseq reads were produced, along with genotypes from the Illumina HumanOmni1-Quad BeadChip. Mapped RNAseq reads for duplicate samples that passed quality control were merged for further analysis, and samples failing QC were discarded.

Geuvadis Project[58]: LCL samples from the 1000 Genomes cohort were RNA sequenced to produce 75 bp paired-end RNAseq reads. Mapped DNA sequence data from phase 1 of the project were downloaded from the 1000 Genomes FTP site (v5a.20130502).

TwinsUK Project[59]: Female monozygotic twin pairs, dizygotic twin pairs and singletons, aged between 38 and 85 were recruited for RNA sequencing and genotyping. Biopsies from subcutaneous adipose tissue and skin were collected, as well as peripheral blood samples for additional generation of lymphoblastoid cell lines (LCLs). 50 bp paired-end RNAseq data were produced from these tissues as well as genotypes from Illumina HumanHap300 and Illumina HumanHap610Q genotyping arrays.

GTEx (Genotype-Tissue Expression) Project[60]: Multiple tissue samples were collected from deceased individuals for RNA sequence analysis and dense genotyping, with sample age range varying between 20 and 71. We use a combination of data from the pilot and midpoint phases of the GTEx project, where samples were genotyped in the Illumina Omni5M and Illumina Omni2.5M genotyping arrays, respectively. RNAseq read lengths produced by the project varied, and we analyse samples with 75-bp long reads only.

Full data accession information, sample sizes and tissue types are described further in Supplementary Table 1. Informed consent was obtained by the original studies and the project was approved by each relevant data access committee.

**RNAseq mapping**. FastQC [v0.11.3] (https://www.bioinformatics.babraham.ac.uk/projects/fastqc/) was run on raw RNAseq data, and samples with drops in base quality below phred 20 or uncalled bases in the middle of reads were discarded. RNAseq reads were then pre-processed to remove adaptor sequences and low quality trailing bases (Phred < 20) using TrimGalore [v0.4.0] (https://www.bioinformatics.babraham.ac.uk/projects/trim_galore/). Poly-A/T sequences >4 bp were also removed from read termini using PRINSEQ-lite [v0.20.4][61]. Remaining reads with >20 nucleotides were then mapped to the human reference sequence (1000G GRCh37 reference, which contains the mitochondrial rCRS NC_012920.1) using STAR [2.5.2a_modified] 2-Pass mapping, allowing ~1 mismatch per 18 bases per read, rounded down to the nearest integer. STAR soft-clipping was also allowed. After mapping, FastQC [v0.11.3] was rerun on data, and samples with median sequence quality scores falling below Phred 20 were removed from further analysis. SAMtools [v1.4.1][62] was then used to retain only properly paired and uniquely mapped reads. This stringent step was applied to ensure that analysed reads originated from the mitochondria, rather than nuclear-encoded fragments of mitochondrial DNA (NUMTs). Transcript abundances were calculated using the 'intersection non-empty model' within HTseq [v0.6.0][63], and gene expression counts were then quantified according to transcripts per million (TPM). Genes expressed in all samples with an average TPM value >2 were used to calculate principal components (PCs) in R and outliers identified from visualisation of PC1, PC2 and PC3 were excluded. Samples were further excluded for having: fewer than 5,000,000 remaining reads, fewer than 10,000 mitochondrial reads, rRNA content >30%, RNAseq mismatch percentage >1%, or intergenic read percentage >30%.

**Quality control, phasing and imputation of genotype data**. QTLtools [v1.0] (https://qtltools.github.io/qtltools/) was used to ensure sample labelling was consistent between genotype and RNAseq data. Quality control (QC) of genotype data was carried out using PLINK [v1.90b3.44][64]. Duplicate samples, genetic PC outliers, samples with unexpected relatedness and samples with outlying heterozygosity rates were removed, in addition to samples with discrepant reported and genotypic sex information, or ambiguous X chromosome homozygosity estimates. Samples with >5% missing genotype data were also excluded. SNPs were removed for violating Hardy–Weinberg equilibrium (HWE) with a *P*-value <0.001, for having a genotype missingness >5% or for having a minor allele frequency (MAF) <1%. SNPs coded according to the negative strand were flipped to the positive strand. SNPs remaining on autosomal chromosomes were phased using default settings within SHAPEIT [v2.r837][65], for all datasets with array genotype data. Phased chromosomes were imputed in 2 Mb intervals using default settings within IMPUTE2 [v2.3.2] using 1000 Genomes Phase 3 individuals as a reference population[66,67]. Imputed genotypes were then hard called with a minimum calling threshold of 0.9 using GTOOL [v0.7.5] (https://www.well.ox.ac.uk/~cfreeman/software/gwas/gtool.html) and filtered out for having an IMPUTE2 info score <0.8, MAF < 5%, genotype missingness > 5%, HWE P < 0.001 or for being multi-allelic.

Datasets genotyped on two different arrays were imputed separately and then merged.

**Quantification of mismatch rate at modified sites**. Previous studies have shown that the proportion of mismatching bases at certain sites on the mitochondrial transcriptome can be used to represent the level of post-transcriptional methylation at these sites. During library preparation for RNA sequencing, methylation modifications on transcripts can interfere with the reverse-transcription process by causing the reverse transcriptase to randomly incorporate nucleotide bases at the methylated position[8]. Though not a direct measurement of methylation level[8], it is thought that this mismatch signature can be used to estimate the level of methylation present on transcripts by measuring the proportion of non-reference alleles at modified sites[14–16,26]. In line with this, the following results support the use of sequence mismatches as a proxy for RNA m1A/G modification level. First, in our previous work we demonstrated that methylation levels estimated via mismatches are repeatable across experiments[14]. To do this we quantified the proportion of sequence mismatches occurring at modified sites using data generated from Illumina sequencing, and then recalculated these proportions for a subset of individuals after performing library preparation and sequencing (from stock blood) on an alternative platform (Ion Torrent), finding that sequence mismatch levels were significantly correlated across platforms (see Supplementary Fig. 2 in Hodgkinson et al.[14], $r^2 = 0.731$, $P = 4.89e−5$). Second, to follow on from this work, we now show that sequence mismatch levels at modified sites are repeatable across independent Illumina sequencing experiments performed on the same samples. To do this, we focussed on a subset of samples that were sequenced two independent times, in each case starting from stock RNA, in the CARTaGENE project (47 samples). For each of these samples we then compared the proportion of mismatches occurring at each modified site (where there is >20× coverage), again finding a high correlation with unique experiments (Supplementary Figure 2, $R = 0.97$, $P < 2.2e−16$, Pearson correlation). Third, Clark et al.[10] made a comparison of samples treated with demethylation enzymes to untreated controls and confirmed the presence of methylation at the ninth position of 19/22 mt-tRNA positions, at similar levels to when measured by primer extension[10]. Although an alternative reverse transcriptase enzyme was used in that study (TGIRT), Safra et al.[11] have shown that sequence mismatch levels inferred when using TGIRT significantly correlate with those inferred when Superscript II (an enzyme used in many RNA sequencing studies) is used during RNAseq library preparation ($r = 0.82$, $P = 3.57e−26$, Pearson correlation, see extended Fig. 1f in that study[11]).

Here, we consistently detect modification levels at levels ≥1% at 13/19 of these mt-tRNA sites, which correspond to the following mitochondrial genomic coordinates: 585, 1610, 3238, 4271, 5520, 7526, 8303, 9999, 10413, 12146, 12274, 14734 and 15896. Of these sites, m1G modifications occur at positions 3238 and 4271, and m1A modifications occur at all other positions[10]. We also calculate methylation level at mtDNA positions 2617 and 13710, which correspond to locations in *mt-RNR2* and *mt-ND5*, respectively; methylation levels at these sites can also be determined using RNAseq data[8,9]. In all cases, m1A/G modifications have been detected at these sites using other high-throughput approaches based around m1A-seq (Supplementary Table 5).

Positional read coverage at mitochondrial sites were summarised using SAMtools mpileup and mismatch rate was calculated from sites with a nucleotide quality score ≥ Phred 13 and coverage ≥ 20×. A site was then considered to show evidence of being methylated if the average proportion of mismatches within a dataset was >1%, as below this level, mismatches due to the presence of methylation is indistinguishable from mismatches due to sequencing error. A combined measure of methylation level was also calculated by averaging across 11 mt-tRNA p9 sites: 585, 1610, 4271, 5520, 7526, 8303, 9999, 10413, 12146, 12274 and 14734 (where values are present), in order to gain an idea of processes influencing post-transcriptional methylation overall. These sites consistently show variation in whole blood data and have previously been used as an estimate of combined methylation[14,15]. Inferred methylation values 3 standard deviations from the mean were masked to avoid association results being driven by extreme values.

Pearson's correlation coefficients between inferred methylation levels at different tRNA P9 sites (including averaged levels across P9 sites) were calculated within individuals, across all datasets and tissue types available. A total of 91 comparisons that were carried out. Pearson's correlation coefficients between methylation level at the same position, across multiple tissues, were carried out using measurements from the GTEx dataset. For a position to be compared between tissues, we required that both tissues have an inferred average methylation level of 1% and at least 100 pairs of data points to compare. For tRNA P9 sites, there are 1657 comparisons, for positions 2617 and 13710 there are 219 comparisons.

**Quantitative trait mapping and meta-analysis**. Quantitative trait mapping was carried out for modification positions with an inferred average methylation level ≥ 1% variation per dataset. Analyses were carried out separately for each position and tissue (therefore comparing samples that were generated using the same library preparation and sequencing protocols), using linear models in PLINK [v1.9][64]. For the TwinsUK tissues, GEMMA [v0.96][68] was used to calculate relatedness matrices and association tests were carried out using univariate linear mixed models. Covariates used in the linear model included 5 study specific genetic PCs and 10

PEER factors calculated from RNAseq data using PEER [v1.0][69] for tissues with ≥100 samples or 5 genetic PCs and 5 PEER factors for tissues with <100 samples. Additional covariates included in the linear model were sex, genotyping array and RNA-sequencing batch information, where available and where relevant. Tissues with multiple datasets were meta-analysed using PLINK [v1.9], under a fixed effects model.

To check for additive effects where mitochondrial RNA methylation levels are associated with two independent nuclear loci having the same direction of effect (five cases), we used CARTaGENE data since these criteria were only met in results from whole blood. For each case we then compared RNA methylation levels for individuals that were heterozygous at the nuclear loci with the highest minor allele frequency and homozygous for the reference allele at the other nuclear loci against RNA methylation levels for individuals that were heterozygous at both sites with a *t*-test.

**Cis-eQTL identification and mediation analysis**. To identify potential genes through which nuclear genetic variants associated with mitochondrial post-transcriptional methylation levels were acting, we carried out mediation analysis. First, for each peak SNP associated with mitochondrial RNA methylation levels, we identified any nuclear genes within a 1 Mb whose expression is significantly associated with the peak nuclear SNP genotype (correcting for the number of *cis*-genes tested for association, per peak SNP), using the same covariates as described above and quantile normalised TPM values from the corresponding tissue (for tissues with multiple independent datasets, we use the dataset with the largest sample size). For nuclear genes/genetic variants that pass these criteria, we then tested whether the expression of the nuclear gene significantly mediated the relationship between the peak nuclear variant and associated inferred mitochondrial RNA methylation levels using 1000 bootstrapping simulations with the 'Mediation' package in R, correcting the *P*-value for the number of *cis*-genes tested.

**Consequences of variation in mitochondrial RNA methylation levels**. To test if methylation levels at mitochondrial tRNA P9 sites have an impact on the expression levels of immediately upstream mitochondrial genes, we regressed the expression levels of the upstream gene on the level of methylation at the relevant tRNA P9 site, including batch, gPC 1–5 and PEER factors 1–10 as covariates in the model. This analysis was carried out in the CARTaGENE whole blood dataset, which is our largest, paired-end, long read dataset, and replicated in the GTEx whole blood dataset.

**Overlap with disease**. We tested for overlap between peak nuclear genetic variants associated with inferred mitochondrial RNA methylation levels, and SNPs in LD ($D' ≥ 0.9$) within a 500 kb interval, with genome-wide significant SNPs ($P < 5E10^{−8}$) associated with disease/disease risk phenotypes reported in the NHGRI-EBI GWAS Catalogue in March 2019[32], requiring the GWAS variants to be identified in studies on European individuals with at least 10,000 cases or quantitative measures. We report $D'$, $R^2$ and MAF values from the largest dataset that is representative of the tissue in which peak SNPs were initially identified.

**Statistics and reproducibility**. Sample sizes are outlined in Supplementary Table 1. Associations between nuclear genetic variants and mitochondrial m1A/G methylation levels were determined via PLINK and GEMMA, and we report asymptotic *P* values generated from the Wald test. Covariates included are listed above per dataset and *p*-values were Bonferroni corrected using the number of methylation sites examined and the number of tissues included in the analyses. Comparisons of mitochondrial m1A/G methylation levels between methylated sites and across tissues were made by calculating Pearson's correlation coefficients. All analyses are reproducible with access to RNA sequencing and genetic data.

**Reporting summary**. Further information on research design is available in the Nature Research Reporting Summary linked to this article.

## Data availability

Raw RNA sequencing and genotype data were obtained from five publicly available projects. For the CARTaGENE project, data were obtained through application to the data access committee (instructions are available at www.cartagene.qc.ca). For the NIMH Genomics Resource, data were obtained after application to the data access committee (through www.nimhgenetics.org). For the Geuvadis Project, RNA sequencing data were obtained from the European Nucleotide Archive under submission number ERA169774, and genetic data were obtained from the 1000 Genomes FTP site (v5a.20130502). For the TwinsUK Project, data were obtained from the European Genome-Phenome archive (https://ega-archive.org) through study ID EGAS00001000805. For the GTEx Project, Data were obtained by application to dbGaP through accession number phs000424.v6.p1. Data underlying each figure is available as Supplementary Data.

## Code availability

Code to infer RNA modification levels from RNA sequencing data is hosted at https://github.com/AJHodgkinson/Mitochondria.

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

## Acknowledgements
We thank the CARTaGENE platform, GTEx, TwinsUK, the NIMH Genomics Resource and the Geuvadis project for use of RNA sequencing data. A.H. holds a Medical Research Council (MRC) eMedLab Medical Bioinformatics Career Development Fellowship, funded from award MR/L016311/1. A.H. also holds a WHRI-Academy Marie Curie (COFUND) Fellowship and the research leading to these results has received funding from the People Programme (Marie Curie Actions) of the European Union's Seventh Framework Programme (FP7/2007-2013) under REA grant agreement no. 608765. Work presented here reflects only the author's views and not the views of the European Commission. A.T.A. is supported by the Generation Trust. Y.I. is funded by a New York University Abu Dhabi research grant (AD105). The research was supported by the National Institute for Health Research (NIHR) Biomedical Research Centre based at Guy's and St Thomas' NHS Foundation Trust and King's College London. The views expressed are those of the authors and not necessarily those of the NHS, the NIHR or the Department of Health. For GTEx data: the Genotype-Tissue Expression (GTEx) Project was supported by the Common Fund of the Office of the Director of the National Institutes of Health (commonfund.nih.gov/GTEx). Additional funds were provided by the NCI, NHGRI, NHLBI, NIDA, NIMH, and NINDS. Donors were enroled at Biospecimen Source Sites funded by NCI\Leidos Biomedical Research, Inc. subcontracts to the National Disease Research Interchange (10XS170), Roswell Park Cancer Institute (10XS171), and Science Care, Inc. (X10S172). The Laboratory, Data Analysis, and Coordinating Center (LDACC) was funded through a contract (HHSN268201000029C) to The Broad Institute, Inc. Biorepository operations were funded through a Leidos Biomedical Research, Inc. subcontract to Van Andel Research Institute (10ST1035). Additional data repository and project management were provided by Leidos Biomedical Research, Inc. (HHSN261200800001E). The Brain Bank was supported supplements to University of Miami grant DA006227. Statistical Methods development grants were made to the University of Geneva (MH090941 & MH101814), the University of Chicago (MH090951, MH090937, MH101825, & MH101820), the University of North Carolina - Chapel Hill (MH090936), North Carolina State University (MH101819), Harvard University (MH090948), Stanford University (MH101782), Washington University (MH101810), and to the University of Pennsylvania (MH101822). For NIMH data: data was provided by Dr. Douglas F. Levinson (dflev@stanford.edu). We gratefully acknowledge the resources were supported by National Institutes of Health/National Institute of Mental Health Grants 5RC2MH089916 (PI: Douglas F. Levinson, M.D.; Co-investigators: Myrna M. Weissman, Ph.D., James B. Potash, M.D., MPH, Daphne Koller, Ph.D., and Alexander E. Urban, Ph.D.) and 3R01MH090941 (Co-investigator: Daphne Koller, Ph.D.). For TwinsUK data: The TwinsUK study was funded by the Wellcome Trust and European Community's Seventh Framework Programme (FP7/2007-2013). The TwinsUK study also receives support from the National Institute for Health Research (NIHR)-funded BioResource, Clinical Research Facility and Biomedical Research Centre based at Guy's and St Thomas' NHS Foundation Trust in partnership with King's College London.

## Author contributions
A.H. and Y.I. designed the study. A.T.A. processed raw data and performed QC and analyses. A.T.A., Y.I. and A.H. wrote the paper.

## Competing interests
The authors declare no competing interests.
