## [Peer Review File · Communications Biology]

Reviewers' comments:

Reviewer #1 (Remarks to the Author):

Ali et al address an interesting question concerning the role of RNA modifications in mitochondrial transcripts, and data which demonstrates control over mt RNA methylation by nuclear genetic variants - some of which are novel.

The authors' approach to identifying m1A modifications by identifying miss-incorporation events is, at this point, largely well supported in the literature, particularly when many sequencing reads or some form of "ultra-deep-sequencing" is used. It does make sense to make use of the many available publically available datasets. Indeed, in this case it is arguable that making use of these helps alleviate bias/batch error, as well as any effects from a single sequencing machine or lab bench. Their subsequent analyses and discussion seem appropriate for their findings. By my understanding their RNA-seq data processing is very stringent, which is encouraging given that fundamental claims are based only on sequencing data.

However, the claim that this work quantifies variation (as opposed to location, as is the common use for RNA-seq in RNA methylation studies) in RNA modification at functionally important positions in the human mitochondrial transcriptome does need some validation of several candidate positions. This should show that their predicted methylation % is corroborated by a further method. Some suggestions for this could include an RNA-seq experiment with an antibody for m1A, a dimroth-reaction experiment showing loss of m1A signal and/or gain of m6A signal in the target site(s), or site-specific 2D TLC (i.e. SCARLET) or mass-spectrometry. Towards this point, reference 13 does not quite support the assertion that is made between lines 113 and 116, but rather the reference posits that RNA post-transcriptional modifications may explain the base mutations observed in sequencing. Considering that Ali et al then use this phenomenon to identify a number of mutations and make important claims (that are indeed interesting), it is important that there is some confirmation of these sites of interest. Ali et al make the point that reference 10 confirmed an RNA-seq approach is suitable for mt RNA (m1A) methylation, but to my understanding the study in reference 10 used TGIRT reverse transcriptase specifically to increase the detectable mutations, I could not find the library preparation strategies and reverse transcriptases used in the data used by Ali et al, but TGIRT is not commonly used so it is likely that data produced with the enzyme would need to be specifically sought out. I do not think that this invalidates the use of the information provided in references 10 and 13, but I do think the further validation in the present study is still required.

The authors should also clarify in the text how they controlled for library preparation methods, as different reverse transcriptases and sequencing strategies will very much affect the sorts of mutations they are attempting to detect. A good orthologous detection method as outlined above could also absolve this issue.

Beyond these concerns I think the paper is well written, and likely of interest to the wider community. The main figures are clear and informative, which should be commended given the difficulty in presenting sequencing data in a clear, concise (and interesting) manner. If the concerns above can be addressed, then the functional annotation of nuclear genetic variation offers further genuine impact that may be useful to those studying mitochondrial biology. It is encouraging that Ali et al replicate a number of previous associations, both of methylated positions, and of genetic variants. Identification of new genetic variants that mediate RNA methylation levels is generally exciting and may have wide-reaching impact.

The use of publicly available data and clearly explained methods makes me confident that a

researcher could reproduce these data.

Reviewer #2 (Remarks to the Author):

The manuscript presents an interesting study of RNA modification m1A from RNA-seq data. The work involved a large amount of RNA-seq data and reported results broadly related to many cell lines/tissues. The presentation is smooth and the structure is clear. Nevertheless, I have the following comments.

My main concern is about the method, i.e., whether it is possible to directly estimate m1A methylation level from RNA-seq data. Considering m1A is a rare modification, it seem to me that the proposed method can be easily affected by many other factors, and the detected m1A sites may have very high false positive rates and the methylation level quantification can be biased. This concern is related to point 1 and 2 in the following.

(1) Please critically compare the m1A sites detected from the proposed method and other high-throughput approaches. How many of the sites detected in this study can or cannot be confirmed by previous studies. Provide illustrated or tabled quantitative results. There are matched cell lines between published studies and the RNA-seq data used in this analysis, and it should be easy to start from there.

(2) Please critically evaluate the accuracy of the quantification. How accurate is the proposed quantification method?

(3) The manuscript focused on m1A methylation, which is not well-studied compared with m6A. It is better to be clear in the title of this manuscript. Maybe change it to: "Nuclear genetic regulation of human mitochondrial m1A RNA modification".

(4) There seems also discussions related to m1G. m1A/G appeared several times. Please provide more references about m1G modification if that is the case.

Reviewer #3 (Remarks to the Author):

Ali et al. analyzed 11,552 samples from 39 tissue/cell types to quantify the variation of three m1A sites in mitochondrial RNA. They identified links between m1A levels and genetic variants in the nuclear genome, and found that these genetic variants are associated with disease phenotypes. Collectively, this work provides candidates that may be involved in the function of m1A, and hints potential physiology function of m1A in mitochondrial RNA. However, there are so many issues and mistakes that should be addressed.

1. This study only focused on the three m1A sites in mitochondrial RNA, but both the title and the abstract do not mention m1A modification. Instead, the authors vaguely used "mitochondrial RNA modification".

2. Why the authors only focused on these three m1A sites in mitochondrial RNA? For instance, m1A at position 58 of mitochondrial tRNAs is also a well-known modification. The authors mentioned that "the

average methylation level at mt-rRNR2 (position 2617) and mt-ND5 (position 13710) transcripts are generally high across all tissues examined”, while methylation levels of tRNA P9 sites appear more dynamic. Are all m1A sites in tRNA more dynamic among different tissue/cell types, or only P9 sites?

3. In abstract, “...find evidence that modification levels impact mitochondrial transcript processing.”. The authors only find the association between modification levels and mitochondrial transcript processing. Please tone down this strong conclusion.

4. In abstract, “...multiple disease phenotypes, including blood pressure, breast cancer...”. Blood pressure is not a disease phenotype.

5. Line 55, “However, these studies are often small in scale and limited to specific cell lines”. The authors should explain why these studies are small in scale, since these technologies can detect modifications on a transcriptome-wide scale and the authors only detected the three m1A sites in mitochondrial RNA. Moreover, in the reference #12 of the manuscript, m1A-seq has been applied to detect m1A methylome in mouse liver, so these studies are not limited to specific cell lines. Furthermore, the authors should include this reference: Li, X. et al. Transcriptome-wide mapping reveals reversible and dynamic N(1)-methyladenosine methylome. *Nat Chem Biol* 12, 311-6 (2016).

6. Line 61, “...detect levels of particular types of RNA modification (m1A/G)...”. Only one type of RNA modification (m1A) has been detected in this study.

7. Line 132 “average levels of 11-25%, 7-12%, 11%, and 10%”, line 149 “Average mt-RNR2 transcript methylation levels range between 38% in GTEx Testis and 67% in GTEx Heart”, line 151 “average levels of transcript methylation”, and so on. Although mismatch rate positively correlates with modification level of m1A, it do not equal to modification level. The authors should modify.

8. What is the difference between Fig.S1 and Fig.S2? If the bar-graph and the box-plot are the same results, please remove Fig.S1. Fig.1a should be displayed in box-plot or violin-plot.

9. The authors did not cite Fig.2.

10. As expected, “an intronic SNP is associated with increased levels of methylation at mt-RNR2, as well as increased expression of TRMT61B”, since methyltransferase TRMT61B catalyzes the formation of m1A in mt-RNR2. However, because TRMT10C is responsible for the formation of m1A at P9 sites in mitochondrial tRNAs, why the authors do not identify TRMT10C?

11. “we identify intronic variants in both whole blood (rs11684695) and subcutaneous adipose (rs10166861) that are significantly associated with methylation levels at position 2617 within mt-RNR2 that mediate these associations through the expression of genes including TRMT61B, a mitochondrial methyltransferase.” Please modify this sentence.

Reviewer #4 (Remarks to the Author):

In this study by Hodgkinson *et al.* entitled “Nuclear genetic regulation of human mitochondrial RNA modification,” the authors identify genetic variants (nuclear-encoded) that are associated with changes in mitochondrial RNA modification as assessed from mutations in RNA-seq data. The extent of work done in this manuscript is admirable with over 11,000 publicly available RNA-seq datasets analyzed. However, it is not clear what the impact of this relatively short, two-figure paper is. What is the general concept learned from this work, that contributes to and perhaps redirects the field. Is this meant to be a resource paper with the list of genetic associations provided in Table 1?

Figure 1 shows that methylation levels can vary between tissues in tRNA, ribosomal RNA and ND5 (the latter of which has been pointed out in a previous study). And while this is certainly good to document, I am not sure this was unexpected. Figure 2 shows examples of genetic variants that associate with the methylation levels of various mitochondrial genes. This is pretty interesting because the variants are in the nuclear genome. However, with almost any quantitative trait that you analyze you will find some variant that associates with it. A little more work needs to be done to demonstrate why these particular associations are important and how they may be regulating methylation.

So in all, this paper is definitely heading in the right direction but I think it is one (possibly two) figure short from telling a complete story. Unless there are some overarching principles to be learned from the computational work that can be incorporated as a concluding figure, wet lab work is seemingly inevitable to conclude this paper. If the authors chose to include a wet lab figure I would recommend taking a few variants localized to genes (perhaps focusing on those associated with diseases), depleting the respective genes by siRNA, and testing whether mitochondrial methylation levels change (with RNA-seq and actual biochemical assays like TLC or primer extension). If the authors are lucky, they may be able to find depletion datasets in GEO to complement their findings. Next, I would investigate how knockdown is affecting methylation. Is it changing levels of the known methyltransferases? If so, this would mean the authors have identified and validated actual regulators of methylation for mitochondrial RNAs. That would make a compelling paper worthy of Nature Communications Biology.

Other concerns:

- Please mention in the title and abstract the exact modification the paper is ultimately investigating (m1A and m1G).
- Also please spell out the modification the first time you use the abbreviation (e.g. N1-methyladenosine).
- Please host the code that was used to analyze this data either in a public repository (e.g. github) or include it with the paper. The methods are detailed but the exact code used needs to be included for reproducibility.

Reviewer report:

Reviewer #1 (Remarks to the Author):

Ali et al address an interesting question concerning the role of RNA modifications in mitochondrial transcripts, and data which demonstrates control over mt RNA methylation by nuclear genetic variants - some of which are novel.

The authors' approach to identifying m1A modifications by identifying miss-incorporation events is, at this point, largely well supported in the literature, particularly when many sequencing reads or some form of "ultra-deep-sequencing" is used. It does make sense to make use of the many available publically available datasets. Indeed, in this case it is arguable that making use of these helps alleviate bias/batch error, as well as any effects from a single sequencing machine or lab bench. Their subsequent analyses and discussion seem appropriate for their findings. By my understanding their RNA-seq data processing is very stringent, which is encouraging given that fundamental claims are based only on sequencing data.

However, the claim that this work quantifies variation (as opposed to location, as is the common use for RNA-seq in RNA methylation studies) in RNA modification at functionally important positions in the human mitochondrial transcriptome does need some validation of several candidate positions. This should show that their predicted methylation % is corroborated by a further method. Some suggestions for this could include an RNA-seq experiment with an antibody for m1A, a dimroth-reaction experiment showing loss of m1A signal and/or gain of m6A signal in the target site(s), or site-specific 2D TLC (i.e. SCARLET) or mass-spectrometry. Towards this point, reference 13 does not quite support the assertion that is made between lines 113 and 116, but rather the reference posits that RNA post-transcriptional modifications may explain the base mutations observed in sequencing. Considering that Ali et al then use this phenomenon to identify a number of mutations and make important claims (that are indeed interesting), it is important that there is some confirmation of these sites of interest. Ali et al make the point that reference 10 confirmed an RNA-seq approach is suitable for mt RNA (m1A) methylation, but to my understanding the study in reference 10 used TGIRT reverse transcriptase specifically to increase the detectable mutations, I could not find the library preparation strategies and reverse transcriptases used in the data used by Ali et al, but TGIRT is not commonly used so it is likely that data produced with the enzyme would need to be specifically sought out. I do not think that this invalidates the use of the information provided in references 10 and 13, but I do think the further validation in the present study is still required.

Reply: We thank the reviewer for their discussion relating to the quantitative nature of our approach, and we believe that it is important to be clear about this in our findings. This question has been at the centre of our thinking since we started working on these questions in 2013. First of all, in order to analyse the scale of data that is required make biological inferences, we make use of pre-existing RNA sequencing datasets that are restricted to this mode of modification quantification (RNA sequencing mismatches). However, given that we use this method for association testing, we believe that it is a robust approach for the analyses and findings we present here. Specifically, our work surveys m1A/G RNA modification levels (as inferred from sequencing mismatches) across individuals within each tissue and compares these to genetic variation in the nuclear genome. In this way, it is the *relative* levels of modification between individuals that are important (that individual A with genotype

XX has a higher level of modification compared to individual B with genotype YY). As such, in order to use sequence mismatches as a proxy for actual RNA modification levels, it is important that the proportion of sequence mismatches at each site is systematic and repeatable, and not driven by noise that causes the measure to vary dramatically across experiments and individuals. To show this, we have performed a number of analyses that confirm that the measure can be replicated across multiple independent scenarios:

1) First, we note that Clark *et al* (RNA 22, 1771-1784, 2016) show that the proportion of mismatches observed at different p9 sites are similar to levels when measured by primer extension, which is an alternative, independent method. The reviewer is correct that this study uses TGIRT (which is rare amongst standard RNAseq libraries), however Safri *et al* (Nature 551, 251-255, 2017) have shown that although TGIRT incorporates mismatches at modified nucleotides at a higher level, the proportion of mismatches observed at modified sites are similar between TGIRT and Superscript II (the enzyme used in the Illumina TruSeq RNA library preparation protocol v2), and significantly correlate ($r=0.82$, $P=3.57e-26$, extended data figure 1F in that publication), thus making data generated using Superscript II valid as a proxy for methylation levels.

2) In our previous work (Reference 13 as discussed above - Hodgkinson, A. *et al. Science* 344: 413-415, 2014), we highlighted that our approach is 'systematic and repeatable across experiments' (lines 113-116 in the original submission) by showing that the proportion of sequence mismatches at each site were highly similar when generated from two different sequencing and library preparation methods. Specifically, within the CARTaGENE project, samples were originally sequenced using the Illumina technology. Then, to confirm that mismatch signals were not occurring at random levels and sites, we repeated RT-PCR reactions and library preparation using stock blood for a number of samples and then sequenced these libraries on the Ion Torrent platform. In doing so, we found that the proportion of mismatches at each p9 site correlated highly between experiments (see Figure S2A in that manuscript for results). We acknowledge that the details of this work were not stated clearly enough in our manuscript and this is now fixed in the revised version (see below).

3) As an extension of this work we now perform new additional analyses to show that sequence mismatch levels are consistent across independent Illumina sequencing experiments. To do this, we focussed on a subset of samples that were sequenced two independent times, in each case starting from stock RNA, in the CARTaGENE project (47 samples). For each of these samples we then compared the proportion of mismatches occurring at each modified site (where there is >20X coverage), again finding a high correlation between unique experiments ($R=0.97$, $P<2.2e-16$).

We acknowledge that these points were not clear in the original version of the manuscript, and so to improve these descriptions, and to add details of the new analyses, we have now substantially expanded the methods section to include this information, starting on page 23, paragraph 2, and as follows:

"Previous studies have shown that the proportion of mismatching bases at certain sites on the mitochondrial transcriptome can be used to represent the level of post-transcriptional methylation at these sites. During library preparation for RNA sequencing, methylation modifications on transcripts can interfere with the reverse transcription process by causing the reverse transcriptase to randomly incorporate

nucleotide bases at the methylated position. Though not a direct measurement of methylation level, it is thought that this mismatch signature can be used to estimate the level of methylation present on transcripts by measuring the proportion of non-reference alleles at modified sites. In line with this, the following results support the use of sequence mismatches as a proxy for RNA m1A/G modification level. First, in our previous work we demonstrated that methylation levels estimated via mismatches are repeatable across experiments. To do this we quantified the proportion of sequence mismatches occurring at modified sites using data generated from Illumina sequencing, and then recalculated these proportions for a subset of individuals after performing library preparation and sequencing (from stock blood) on an alternative platform (Ion Torrent), finding that sequence mismatch levels were significantly correlated across platforms (see Figure S2 in Hodgkinson *et al* (2014), $r^2 = 0.731$, $P = 4.89e-5$). Second, to follow on from this work, we now show that sequence mismatch levels at modified sites are repeatable across independent Illumina sequencing experiments performed on the same samples. To do this, we focussed on a subset of samples that were sequenced two independent times, in each case starting from stock RNA, in the CARTaGENE project (47 samples). For each of these samples we then compared the proportion of mismatches occurring at each modified site (where there is >20X coverage), again finding a high correlation between unique experiments (Supplementary Figure 2, $R=0.97$, $P<2.2e-16$). Third, Clark *et al* (2016) made a comparison of samples treated with demethylation enzymes to untreated controls and confirmed the presence of methylation at the ninth position of 19/22 mt-tRNA positions, at similar levels to when measured by primer extension. Although an alternative reverse transcriptase enzyme was used in that study (TGIRT), Safra *et al* (2016) have shown that sequence mismatch levels inferred when using TGIRT significantly correlate with those inferred when Superscript II (an enzyme used in many RNA sequencing studies) is used during RNAseq library preparation ($r=0.82$, $P=3.57e-26$, see extended figure 1F in that study)."

Beyond this, it is clear that the use of sequencing mismatches in RNA sequencing data as a signal for RNA modification levels is semi-quantitative by nature, relying on misincorporation by the reverse transcription enzyme during library preparation at nucleotides that have been modified. This process is not perfect, meaning that the proportion of mismatches is often lower than the actual level of RNA modification at the site (Li, X. *et al.* Mol Cell 68, 993-1005. 2017). However, alternative methods for quantification of m1A/G RNA modification levels are often limited by not being site specific (and instead measuring modification levels in total across all modified sites within total RNA or a selected transcript), by lack of testing for the type of modification considered here (m1A/G), and by lack of testing in mitochondrial sequences that are highly modified. It is therefore unclear as to whether these approaches would accurately measure the underlying modification rate in the context of the present study, and they would require further testing and validation in themselves. Of the suggested techniques we note that:

1. RNAseq experiments with an antibody for m1A can be used to quantify m1A modification levels at individual sites (as in Dominissini, D. *et al.* Nature 530, 441-446. 2016), but only if a single modification occurs per RNA transcript. In our case, multiple m1A modifications are documented for each MT-tRNA, MTND5 and MTRNR2, and so this approach would not be site specific.
2. A dimroth-reaction experiment to convert m1A to m6A would not be quantifiable and is instead used to determine if a site is modified at the RNA level

(by comparing converted versus non-converted RNAseq data, as in Dominissini, D. *et al.* *Nature* 530, 441-446. 2016 and Li, X. *et al.* *Nat Chem Biol* 12, 311-316. 2016). In line with this, we note that all of the sites we consider in this study have been confirmed to be modified in at least one other high-throughput sequencing study (see reply to reviewer 2, point 1).

3. Mass spectrometry can be used to quantify combined m1A levels across all input material, but since RNA fragments are broken down to nucleosides and short oligonucleotides, it is currently difficult to map back modification levels to single specific sites, particularly in regions that contain multiple modifications sites/types like mt-tRNAs. Li, X. *et al.* (*Nat Chem Biol* 12, 311-316. 2016) apply this approach to look at m1A modifications in human mRNAs, finding levels of ~0.02% across different cell lines, but these are global rates rather than site specific. There has been some work on mapping back modifications to specific sites (e.g. Ross R, *et al.* *Methods*. 107: 73-8. 2016), but these approaches are still in their infancy and have not been thoroughly tested across modification types and genomic regions.
4. SCARLET has been used to quantify levels of m6A modifications at specific sites in a number of different cell lines (e.g. Liu *et al.*, *RNA* 19:1848-1856. 2013), however to our knowledge this approach has never been tested for m1A/G modifications, or in mitochondrial RNA.

We agree with the reviewer that the use of sequencing mismatches as a proxy for RNA modification rates, as well as potential alternative approaches, is an important point that requires further discussion. As such, we now also include a dissection of these issues in the discussion section on page 19, paragraph 1, and as follows:

“Finally, our approach makes use of sequence mismatches in RNA sequencing data as a proxy for m1A/G RNA modification levels. This approach is semi-quantitative by nature since misincorporation of nucleotides by the reverse-transcription enzyme at modified sites during RNA sequencing library preparation is not infallible (see methods). However, our approach allows us to survey rates of RNA modification across a large number of individuals and tissues, drawing power to make important biological inferences from these data. It is hoped in the future that other high-throughput approaches (such as Nanopore sequencing or site-specific mass spectrometry) may have the potential to more directly quantify RNA modification levels across a wider range of modification types, and thus allow for an expanded view on genetic drivers and downstream consequences of dynamic RNA modification regulation.”

In conclusion, given the evidence and additional analyses that we have outlined above (and now include in the manuscript), we believe that our approach is appropriate for our findings and conclusions, and that it is beyond the scope and our expertise to develop and test alternative methods to quantify levels of modification at individual nucleotides.

The authors should also clarify in the text how they controlled for library preparation methods, as different reverse transcriptases and sequencing strategies will very much affect the sorts of mutations they are attempting to detect. A good orthologous detection method as outlined above could also absolve this issue.

Reply: The reviewer is correct that alternative library preparation and sequencing methods may have an effect on our approach, however in this case all data across the

five different sequencing projects used here were generated using the Illumina Truseq protocol for library preparation, and then sequenced on Illumina HiSeq 2000/2500 machines. Beyond this, to ensure that project specific features were not influencing our results, we performed quantitative trait loci mapping within each dataset and tissue separately. By doing this, the associations we identified were as a result of comparing individuals that were sequenced and genotyped using the same methods. For tissues/cell types with multiple independent datasets, we then combined these data within a meta-analysis to calculate the final p-value and beta coefficient; an approach that is robust to dataset specific effects. To make this clearer in the main text, we have now added the following to the methods section on page 25, paragraph 3:

“Analyses were carried out separately for each position and tissue (therefore comparing samples that were generated using the same library preparation and sequencing protocols), using linear models in PLINK [v1.9]”

Beyond these concerns I think the paper is well written, and likely of interest to the wider community. The main figures are clear and informative, which should be commended given the difficulty in presenting sequencing data in a clear, concise (and interesting) manner. If the concerns above can be addressed, then the functional annotation of nuclear genetic variation offers further genuine impact that may be useful to those studying mitochondrial biology. It is encouraging that Ali et al replicate a number of previous associations, both of methylated positions, and of genetic variants. Identification of new genetic variants that mediate RNA methylation levels is generally exciting and may have wide-reaching impact.

The use of publicly available data and clearly explained methods makes me confident that a researcher could reproduce these data.

Reviewer #2 (Remarks to the Author):

The manuscript presents an interesting study of RNA modification m1A from RNA-seq data. The work involved a large amount of RNA-seq data and reported results broadly related to many cell lines/tissues. The presentation is smooth and the structure is clear. Nevertheless, I have the following comments.

My main concern is about the method, i.e., whether it is possible to directly estimate m1A methylation level from RNA-seq data. Considering m1A is a rare modification, it seem to me that the proposed method can be easily affected by many other factors, and the detected m1A sites may have very high false positive rates and the methylation level quantification can be biased. This concern is related to point 1 and 2 in the following.

(1) Please critically compare the m1A sites detected from the proposed method and other high-throughput approaches. How many of the sites detected in this study can or cannot be confirmed by previous studies. Provide illustrated or tabled quantitative results. There are matched cell lines between published studies and the RNA-seq data used in this analysis, and it should be easy to start from there.

Reply: First, it is important to note that the goal of the study was not to detect novel m1A/G sites, but rather to focus on sites that are already known to be modified at the RNA level. However, we have now compared the sites detected here to those that have presented mitochondrial data within m1A-seq approaches (three studies: Safra *et al.*

Nature 551, 251-255. 2017; Clark et al. RNA 22, 1771-1784, 2016; Li, X. et al. Mol Cell 68, 993-1005. 2017). In total, of the 15 sites we consider along the mitochondrial genome (13 P9 sites within different tRNAs, one site in MTRNR2 and one site in MTND5), all have been detected in at least one of these studies, and most of them have been detected in all three studies. To document this in the manuscript, we have now compiled a supplementary table containing this information (Supplementary Table 5), and refer to this table in the methods section on page 24, paragraph 2 as follows:

“In all cases, m1A/G modifications have been detected at these sites using other high-throughput approaches based around m1A-seq (Supplementary Table 5).”

The studies detailed above all use HEK293T cells, and so do not directly match the cell lines and primary tissues that we survey in this study, however given the consistency of sites detected across all approaches, it seems reasonable to expect the same sites would be detected in most cell types.

(2) Please critically evaluate the accuracy of the quantification. How accurate is the proposed quantification method?

Reply: Please see our first reply to reviewer #1 above.

(3) The manuscript focused on m1A methylation, which is not well-studied compared with m6A. It is better to be clear in the title of this manuscript. Maybe change it to: "Nuclear genetic regulation of human mitochondrial m1A RNA modification".

Reply: Since we consider both m1A and m1G RNA modifications, we have changed the title to: “Nuclear genetic regulation of human mitochondrial m1A/G RNA modification”.

(4) There seems also discussions related to m1G. m1A/G appeared several times. Please provide more references about m1G modification if that is the case.

Reply: At p9 sites (the 9th position of tRNAs), we quantify RNA modification levels in 13 different tRNAs along the mitochondrial transcriptome. Of these, two occur at positions where the reference nucleotide is guanine (3238 and 4271), and it was shown in the Clark *et al* study (2016) that m1G modifications occur at these sites. To provide more information and references around the two types of modifications we consider, we have added the following to page 24, paragraph 2:

“Here, we consistently detect modification levels at levels $\geq 1\%$ at 13/19 of these mt-tRNA sites, which correspond to the following mitochondrial genomic coordinates: 585, 1610, 3238, 4271, 5520, 7526, 8303, 9999, 10413, 12146, 12274, 14734 and 15896. Of these sites, m1G modifications occur at positions 3238 and 4271, and m1A modifications occur at all other positions”.

Reviewer #3 (Remarks to the Author):

Ali et al. analyzed 11,552 samples from 39 tissue/cell types to quantify the variation of three m1A sites in mitochondrial RNA. They identified links between m1A levels and genetic variants in the nuclear genome, and found that these genetic variants are associated with disease phenotypes. Collectively, this work provides candidates that may be involved in the

function of m1A, and hints potential physiology function of m1A in mitochondrial RNA. However, there are so many issues and mistakes that should be addressed.

1. This study only focused on the three m1A sites in mitochondrial RNA, but both the title and the abstract do not mention m1A modification. Instead, the authors vaguely used “mitochondrial RNA modification”.

Reply: We have changed the title to: “Nuclear genetic regulation of human mitochondrial m1A/G RNA modification”. We have now also included the specific modification types that we consider in the abstract:

“Here, we quantify variation in m1A/G RNA modification levels at functionally important positions in the human mitochondrial genome across 11,552 samples from 39 tissue/cell types”.

2. Why the authors only focused on these three m1A sites in mitochondrial RNA? For instance, m1A at position 58 of mitochondrial tRNAs is also a well-known modification. The authors mentioned that “the average methylation level at mt-rRNR2 (position 2617) and mt-ND5 (position 13710) transcripts are generally high across all tissues examined”, while methylation levels of tRNA P9 sites appear more dynamic. Are all m1A sites in tRNA more dynamic among different tissue/cell types, or only P9 sites?

Reply: In total we quantify RNA modification levels (via RNA sequencing mismatches) at 15 different sites along the mitochondrial transcriptome. Two of these are in ND5 and RNR2 (both m1A), and the remaining 13 are at the ninth position of different tRNAs (p9 sites – both m1A and m1G modifications). We focus on these sites as they are the only ones that are reliably detectable from RNA sequencing data that has been generated using standard library preparation techniques. Mitochondrial tRNAs are highly post-transcriptionally modified, and it is thought that as many as nine sites per tRNA undergo some form of post-transcriptional methylation (Powell et al. *Front Genet* 2015, 6:79). However, it is not expected that all types of methylation will be detectable in RNA sequencing data for two reasons: 1) The addition of a methyl group to different nitrogen positions in the nucleotide appear to have different effects on the RT enzyme and are therefore not always detectable (Ryvkin et al. *RNA* 2013, 19:1684-1692). 2) Methylation events occur at different stages of tRNA maturation. P9 events are thought to be one of the first modifications to occur (Helm et al. *J Mol Biol* 2004, 337:545-560) and since standard RNA sequencing libraries tend to capture a large fraction of unprocessed/partially processed mitochondrial transcripts, it is likely that earlier events show the strongest signals in this type of data. Since we do not survey any other m1A sites within tRNAs, we cannot comment on how variable they might be across tissues.

3. In abstract, “...find evidence that modification levels impact mitochondrial transcript processing”. The authors only find the association between modification levels and mitochondrial transcript processing. Please tone down this strong conclusion.

Reply: We have now made the following change to the abstract:

“and find evidence that modification levels are associated with mitochondrial transcript processing”

4. In abstract, "...multiple disease phenotypes, including blood pressure, breast cancer...". Blood pressure is not a disease phenotype.

Reply: We agree that this original language is not accurate, and have now made the following change to the abstract:

"Genetic variants linked to RNA modification levels are associated with multiple disease and disease-related phenotypes, including blood pressure, breast cancer, glaucoma and psoriasis".

5. Line 55, "However, these studies are often small in scale and limited to specific cell lines". The authors should explain why these studies are small in scale, since these technologies can detect modifications on a transcriptome-wide scale and the authors only detected the three m1A sites in mitochondrial RNA. Moreover, in the reference #12 of the manuscript, m1A-seq has been applied to detect m1A methylome in mouse liver, so these studies are not limited to specific cell lines. Furthermore, the authors should include this reference: Li, X. et al. Transcriptome-wide mapping reveals reversible and dynamic N(1)-methyladenosine methylome. *Nat Chem Biol* 12, 311-6 (2016).

Reply: We apologise for not being clear about our meaning here. By 'small in size' we meant that studies have not considered many samples, rather than only considering a small number of modified sites. We also meant to convey that most, but not all, of these studies use cell lines rather than primary tissue. To make these points clearer, we have now changed the text to the following on page 3, paragraph 2, and have also included the suggested reference:

"However, these studies often consider a small number of samples mostly limited to specific cell lines, and frequently focus on the detection of novel modification sites rather than attempt to survey the dynamic range of modification level across a large number of individuals".

6. Line 61, "...detect levels of particular types of RNA modification (m1A/G)...". Only one type of RNA modification (m1A) has been detected in this study.

Reply: At p9 sites (the 9th position of tRNAs), we quantify RNA modification levels in 13 different tRNAs along the mitochondrial transcriptome. Of these, two occur at positions where the reference nucleotide is guanine (3238 and 4271), and it was shown in the Clark *et al* study (Clark et al. *RNA* 22, 1771-1784, 2016) that m1G modifications occur at these sites. To make this clearer, we have now amended the text to the following, in the methods section on page 24, paragraph 2:

"Here, we consistently detect modification levels at levels $\geq 1\%$ at 13/19 of these mt-tRNA sites, which correspond to the following mitochondrial genomic coordinates: 585, 1610, 3238, 4271, 5520, 7526, 8303, 9999, 10413, 12146, 12274, 14734 and 15896. Of these sites, m1G modifications occur at positions 3238 and 4271, and m1A modifications occur at all other positions."

7. Line 132 "average levels of 11-25%, 7-12%, 11%, and 10%", line 149 "Average mt-RNR2 transcript methylation levels range between 38% in GTEx Testis and 67% in GTEx Heart", line

151 “average levels of transcript methylation”, and so on. Although mismatch rate positively correlates with modification level of m1A, it do not equal to modification level. The authors should modify.

Reply: We agree that the use of the term ‘RNA methylation levels’ is not accurate in this context, and have now changed this to ‘inferred RNA methylation levels’ throughout the document when we refer directly to values obtained via RNA sequencing mismatches.

8. What is the difference between Fig.S1 and Fig.S2? If the bar-graph and the box-plot are the same results, please remove Fig.S1. Fig.1a should be displayed in box-plot or violin-plot.

Reply: Figures S1 and S2 show very similar results and therefore we have removed Figure S1 as suggested. Fig 1a shows mean methylation levels across tissues, thus highlighting variation on this scale. We have now added bars to these columns to show the standard deviation of each distribution.

9. The authors did not cite Fig.2.

Reply: We apologise for this oversight and have now included a citation to Figure 2 in the results section on page 9, paragraph 1.

10. As expected, “an intronic SNP is associated with increased levels of methylation at mt-RNR2, as well as increased expression of TRMT61B”, since methyltransferase TRMT61B catalyzes the formation of m1A in mt-RNR2. However, because TRMT10C is responsible for the formation of m1A at P9 sites in mitochondrial tRNAs, why the authors do not identify TRMT10C?

Reply: The reviewer is correct that TRMT10C is responsible for the formation of m1A at P9 sites, and there are many possible reasons why we didn’t detect a direct association to this gene in our quantitative trait mapping analysis. Primarily, there would need to be a genetic variant that has a large enough effect on the function or expression of TRMT10C in our data, and this variant would need to have high enough allelic variation across the population for us to detect a link to m1A levels. In the current study, we note that we find only a single functional (missense) variant in MRPP1 above minor allele frequency of 5%. We do note however that we find a strong association with a missense genetic variant within MRPP3 (a gene that forms part of the RNase P complex, along with TRMT10C), which may either directly influence RNA methylation levels at p9 sites, or modulate processes linked to methylation through its involvement in RNase P.

11. “we identify intronic variants in both whole blood (rs11684695) and subcutaneous adipose (rs10166861) that are significantly associated with methylation levels at position 2617 within mt-RNR2 that mediate these associations through the expression of genes including TRMT61B, a mitochondrial methyltransferase.” Please modify this sentence.

Reply: We agree that this sentence is not clear, and have now changed it to the following, on page 11, paragraph 2:

“Finally, we identify intronic variants (rs11684695 and rs10166861) that are significantly associated with inferred methylation levels at position 2617 within *mt-RNR2* (in whole blood and subcutaneous adipose respectively) that mediate these

associations through the expression of genes including *TRMT61B*, a mitochondrial methyltransferase”

Reviewer #4 (Remarks to the Author):

In this study by Hodgkinson et al. entitled “Nuclear genetic regulation of human mitochondrial RNA modification,” the authors identify genetic variants (nuclear- encoded) that are associated with changes in mitochondrial RNA modification as assessed from mutations in RNA-seq data. The extent of work done in this manuscript is admirable with over 11,000 publicly available RNA-seq datasets analyzed. However, it is not clear what the impact of this relatively short, two-figure paper is. What is the general concept learned from this work, that contributes to and perhaps redirects the field. Is this meant to be a resource paper with the list of genetic associations provided in Table 1? Figure 1 shows that methylation levels can vary between tissues in tRNA, ribosomal RNA and ND5 (the latter of which has been pointed out in a previous study). And while this is certainly good to document, I am not sure this was unexpected. Figure 2 shows examples of genetic variants that associate with the methylation levels of various mitochondrial genes. This is pretty interesting because the variants are in the nuclear genome. However, with almost any quantitative trait that you analyze you will find some variant that associates with it. A little more work needs to be done to demonstrate why these particular associations are important and how they may be regulating methylation.

Reply: We believe that our work builds a clear picture of how RNA modification levels vary across individuals and tissues, gives key insight into the possible mechanisms underlying these processes, and draws important conclusions about the downstream consequences of variation in RNA modification levels. Specifically, we report the following novel findings: 1) Mitochondrial RNA modification levels show both diverse and tissue specific patterns across different individuals, 2) Mitochondrial RNA modification levels are linked to changes in mitochondrial gene expression, suggesting a role for these processes in regulating the expression of core components of the cell’s energy generating apparatus, 3) By analyzing the functional consequences of nuclear genetic mutations associated with mitochondrial RNA modification we infer novel roles for nuclear genes (e.g. *PNPT1* and *LONP1*) in mitochondrial post-transcriptional regulation and 4) Nuclear genetic variants associated with mitochondrial RNA modification levels are genetically linked to GWAS SNPs (including blood pressure, breast cancer, glaucoma and psoriasis), thus implicating an under-explored role for variation in mitochondrial RNA modification in complex disease.

So in all, this paper is definitely heading in the right direction but I think it is one (possibly two) figure short from telling a complete story. Unless there are some overarching principles to be learned from the computational work that can be incorporated as a concluding figure, wet lab work is seemingly inevitable to conclude this paper. If the authors chose to include a wet lab figure I would recommend taking a few variants localized to genes (perhaps focusing on those associated with diseases), depleting the respective genes by siRNA, and testing whether mitochondrial methylation levels change (with RNA-seq and actual biochemical assays like TLC or primer extension). If the authors are lucky, they may be able to find depletion datasets in GEO to complement their findings. Next, I would investigate how knockdown is affecting methylation. Is it changing levels of the known methyltransferases? If so, this would mean the authors have identified and validated actual regulators of methylation

for mitochondrial RNAs. That would make a compelling paper worthy of Nature Communications Biology.

Reply: As requested, we have now performed additional computational analyses to better understand a key feature of our study: the underlying genetic architecture of variable mitochondrial m1A/G RNA modification levels. Specifically, for sites in mitochondrial RNA where we identify two independent nuclear loci associated with methylation levels with the same direction of effect, we tested whether independent alleles have an additive effect. Under these criteria we consider five methylated positions in whole blood data, and in all of these cases, we observe a significant change in methylation levels associated with carrying two effect alleles (one at each independent loci) versus carrying only one (Figure 3, $P < 0.05$ after Bonferroni correction). We have now documented this finding in the results section on page 9, paragraph 2 as follows:

“To further consider the underlying genetic architecture of variable mitochondrial RNA methylation levels across individuals, for sites in mitochondrial RNA where we identify two independent nuclear loci associated with methylation levels with the same direction of effect, we tested whether independent alleles have an additive effect. Under these criteria we consider five methylated positions in whole blood data, and in all of these cases, we observe a significant change in inferred methylation levels associated with carrying two effect alleles (one at each independent loci) versus carrying only one (Figure 3, $P < 0.05$ after Bonferroni correction).”

As well as in the methods section on page 26, paragraph 2 as follows:

“To check for additive effects where mitochondrial RNA methylation levels are associated with two independent nuclear loci having the same direction of effect (five cases), we used CARTaGENE data since these criteria were only met in results from whole blood. For each case we then compared RNA methylation levels for individuals that were heterozygous at the nuclear loci with the highest minor allele frequency and homozygous for the reference allele at the other nuclear loci against RNA methylation levels for individuals that were heterozygous at both sites with a t-test.”

Other concerns:

-Please mention in the title and abstract the exact modification the paper is ultimately investigating (m1A and m1G).

Reply: We have changed the title to: “Nuclear genetic regulation of human mitochondrial m1A/G RNA modification”. We have now also included the specific modification types that we consider in the abstract:

“Here, we quantify variation in m1A/G RNA modification levels at functionally important positions in the human mitochondrial genome across 11,552 samples from 39 tissue/cell types”.

-Also please spell out the modification the first time you use the abbreviation (e.g. N1-methyladenosine).

Reply: We have now included the full name of the modifications the first time that they are mentioned in the main text on page 4, paragraph 1.

-Please host the code that was used to analyze this data either in a public repository (e.g. github) or include it with the paper. The methods are detailed but the exact code used needs to be included for reproducibility.

Reply: We have now added code to GitHub that computes modification levels from sequencing data in the same way as described in this study. This information has been added to the manuscript in the acknowledgements section on page 29, paragraph 1.

REVIEWERS' COMMENTS:

Reviewer #1 (Remarks to the Author):

Having read the revised manuscript and rebuttal letter. I feel that my concerns have been addressed. Particularly in the expansion of the methods sections to describe how the analyses were performed and clarifying the data and controls used.

The additions to the discussion also clarify the utility of their approach whilst allowing the reader to understand the potential shortcomings in the methods and data.

Given the responses and additions to the paper, I do think it is beyond the required scope to develop and test alternative methods to quantify the modifications at individual nucleotides.

With these concerns addressed I think the paper is suitable for publication - I enjoyed reading it. The approach is thoughtful and well considered. The comparisons and controls for the data do back up the conclusions and are now well described in the paper. Importantly, I think the figures are well presented and clear to a "non-bioinformatic" audience.

Reviewer #2 (Remarks to the Author):

My main concern was only partially addressed, which may limit the impact and reliability of the study. In general, I still hold similar concern with reviewer #1 during the first round of review.

Reviewer #4 (Remarks to the Author):

This work should be fairly reproducible given the code that has now been included with the study through Github.

From the attached file

The authors of the study "Nuclear genetic regulation of human mitochondrial RNA modification" have satisfactorily responded to all minor concerns from the initial critique of the study. These included editing the title, spelling out nucleotide modifications and hosting all the code used for this project in Github.

As for the major concerns the authors have made a convincing case for the contribution of this work to understanding the regulation and diversity of RNA modifications. There are some components that are not entirely novel (such as the tissue specific levels of modifications which for example was demonstrated by Safra et al. (doi: 10.1038/nature24456)) but this study nonetheless reinforces some of these concepts.

While I do appreciate the newly added figure (Fig. 3) I do not think it makes for a strong concluding /take home figure and it does not necessarily make an overarching summary of the work. While this does not necessarily doom the paper, the authors may want to consider perhaps concluding this study with a figure that summarizes the paper instead.